# Unleashing Reasoning Capability of LLMs via Scalable Question Synthesis from Scratch

## Abstract

The availability of high-quality data is one of the most important factors in improving the reasoning capability of LLMs. Existing works have demonstrated the effectiveness of creating more instruction data from seed questions or knowledge bases. Recent research indicates that continually scaling up data synthesis from strong models (e.g., GPT-4) can further elicit reasoning performance. Though promising, the open-sourced community still lacks high-quality data at scale and scalable data synthesis methods with affordable costs. To address this, we introduce ScaleQuest, a scalable and novel data synthesis method that utilizes "small-size" (e.g., 7B) open-source models to generate questions from scratch without the need for seed data with complex augmentation constraints. With the efficient ScaleQuest, we automatically constructed a mathematical reasoning dataset consisting of 1 million problem-solution pairs, which are more effective than existing open-sourced datasets. It can universally increase the performance of mainstream open-source models (i.e., Mistral, Llama3, DeepSeekMath, and Qwen2-Math) by achieving 29.2% to 46.4% gains on MATH. Notably, simply fine-tuning the Qwen2-Math-7B-Base model with our dataset can even surpass Qwen2-Math-7B-Instruct, a strong and well-aligned model on closed-source data, and proprietary models such as GPT-4-Turbo and Claude-3.5 Sonnet.[1]

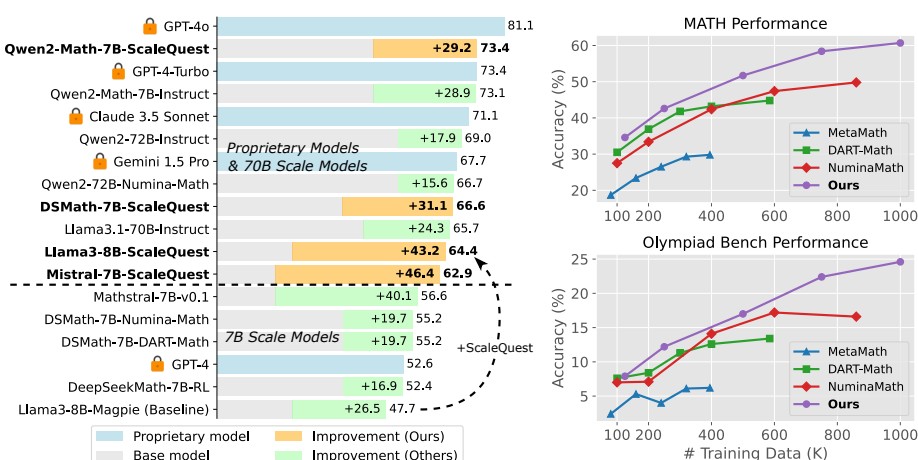

Figure 1: **Left:** Results of different models on MATH, where **-ScaleQuest** denotes ours. **Right:** Results of Llama3-8B fine-tuned on publicly available datasets constructed by different methods.

## 1 Introduction

How to improve the reasoning capabilities of Large Language Models (LLMs) has attracted significant attention. The success of recent advanced models, such as OpenAI o1 and Claude-3.5, heavily depends on access to extensive, diverse, and high-quality reasoning datasets. However, the

---

[1]Code, data, and models will be publicly available.

proprietary nature of the data presents a significant barrier to the open-source community. Recent works have highlighted data synthesis as a promising approach (Ntoutsi et al., 2020) to address data scarcity for instruction tuning (Inan et al., 2023). As recent works have disclosed that crafting the right questions is crucial for eliciting the reasoning capabilities of LLMs (Yu et al., 2023a; Shah et al., 2024), the core of reasoning data synthesis lies in creating large-scale and novel questions.

Previous efforts in reasoning data synthesis have demonstrated the effectiveness of leveraging powerful language models to generate instructions. We categorize these approaches into two types: question-driven approaches and knowledge-driven approaches. Question-driven methods include question rephrasing (Yu et al., 2023a), evol-instruct (Xu et al., 2023; Luo et al., 2023; Zeng et al., 2024), question back-translation (Lu et al., 2024), or providing few-shot examples (Mitra et al., 2024). These methods are limited in data diversity, as the generated problems closely resemble the seed questions, with only minor modifications such as added conditions or numerical changes. This lack of diversity hampers their scalability potential. To improve question diversity, recent knowledge-driven works (Huang et al., 2024b) scale question synthesis by constructing knowledge bases (Li et al., 2024b) or concept graphs (Tang et al., 2024) and sampling key points (Huang et al., 2024a) from them to generate new questions. Nevertheless, the above two types of approaches commonly rely on strong models, like GPT-4, to synthesize new questions, but the high API costs make it impractical to generate large-scale data. As a result, despite these advancements, the open-source community still faces a shortage of high-quality data at scale and cost-effective synthesis methods.

To meet this requirement, we explore a scalable, low-cost method for data synthesis. We observe that using problem-solving models to directly synthesize reasoning questions, as explored in Yu et al. (2023b) and Xu et al. (2024), falls short in synthesizing reasoning data, as shown in Figure 1 (see Llama3-8B-Magpie results). Accordingly, we propose a novel, scalable, and cost-effective data synthesis method, ScaleQuest, which first introduces a two-stage question-tuning process consisting of Question Fine-Tuning (QFT) and Question Preference Optimization (QPO) to unlock the question generation capability of problem-solving models. Once fine-tuned, these models can then generate diverse questions by sampling from a broad search space without the need for additional seed questions or knowledge constraints. The generated questions can be further refined through a filtering process, focusing on language clarity, solvability, and appropriate difficulty. Moreover, we introduce an extra reward-based filtering strategy to select high-quality responses.

We generated data based on two lightweight, open-source models: DeepSeekMath-7B-RL (Shao et al., 2024) and Qwen2-Math-7B-Instruct (Yang et al., 2024a), producing a final dataset of 1 million question-answer pairs. As shown in Figure 1, our synthetic dataset boosts performance by 29.2% to 46.4% across four major open-source models: Mistral-7B (Jiang et al., 2023), Llama3-8B (Dubey et al., 2024), DeepSeekMath-7B (Shao et al., 2024), and Qwen2-Math-7B (Yang et al., 2024a). Compared with other publicly available datasets such as MetaMath (Yu et al., 2023a), DART-Math (Tong et al., 2024), and NuminaMath (Li et al., 2024c), our approach demonstrates great scalability in both in-domain and out-of-domain evaluation. In terms of in-domain evaluation, our method outperforms existing high-quality open-source datasets, achieving better results with the same amount of data. For out-of-domain evaluation, compared with other datasets, the performance of our synthetic dataset continues to show promising trends as the volume of training data increases, indicating significant potential for further improvements through ongoing data scaling.

## 2 SCALEQUEST: SCALING QUESTION SYNTHESIS FROM SCRATCH

In this section, we first explain the motivation and process of our question generation method (section 2.1). Then, we introduce how to train a question generator via Question Fine-Tuning (section 2.2) and Question Preference Optimization (section 2.3). Next, we use the question generator to generate math questions, followed by a filtering process (section 2.4). Finally, we describe the response generation process (section 2.5). The overview of our method is illustrated in Figure 2.

### 2.1 QUESTION GENERATION FROM SCRATCH

The question generation process involves providing only a few prefix tokens from an instruction template (e.g., "`<|begin_of_sentence|>User:`") to guide the model in question generation. A fine-tuned causal language model, which has learned to generate responses

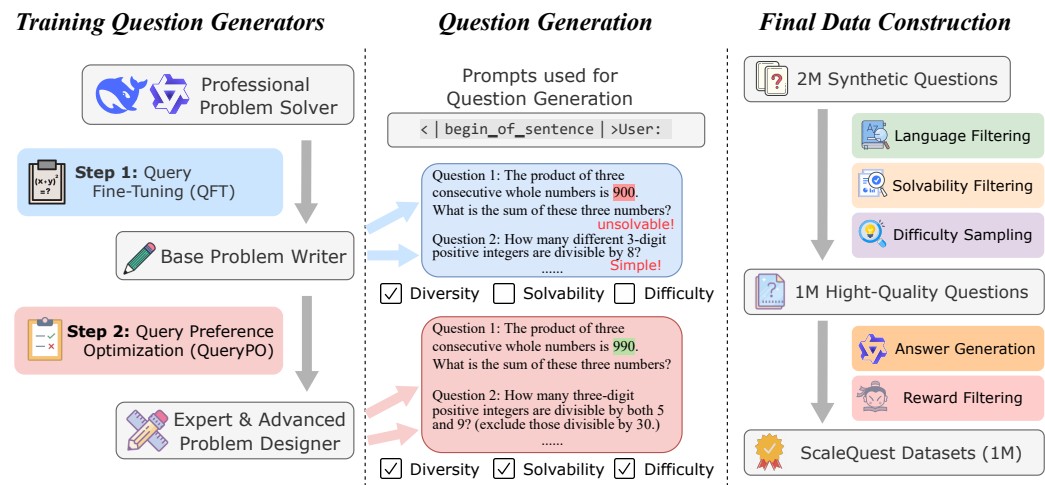

Figure 2: Overview of our ScaleQuest method.

based on question-answer pairs (e.g., "`<|begin_of_sentence|>User: {Question}. Assistant: {Response}`"), could potentially be leveraged to generate questions directly (Xu et al., 2024). This is because, during instruction tuning, the model is trained using a causal mask, where each token only attends to preceding tokens. This ensures that the hidden states evolve based on past context without future token influence. However, during instruction tuning, the actual loss is calculated based on the response, i.e.,

$$\mathcal{L} = -\log P(y_i|X, y_{<i}), \tag{1}$$

where $X = \{x_1, x_2, \ldots, x_m\}$ denotes question and $Y = \{y_1, y_2, \ldots, y_n\}$ denotes response. Since $P(x_i|x_{<i})$ is inherently modeled, we need to activate the model's capability for question generation.

## 2.2 QUESTION FINE-TUNING (QFT)

To activate the model's question generation capability, we first perform Question Fine-Tuning (QFT), where we train the problem-solving model using a small set of problems. To ensure that the generator stops after producing the questions and does not continue generating a response, we added an end-of-sentence token at the end of each question. We used approximately 15K problems (without solutions) by mixing the training set of GSM8K (Cobbe et al., 2021) and MATH (Hendrycks et al., 2021) datasets as training samples. We train DeepSeekMath-7B-RL Shao et al. (2024) and Qwen2-Math-7B-Instruct Yang et al. (2024a) with these samples.

The purpose of utilizing these problems is to activate the model's question-generation capability rather than to make the model memorize them. To validate this hypothesis, we trained the model separately using the GSM8K and MATH datasets and compared whether the distribution of the generated questions matched that of the training data. To evaluate the

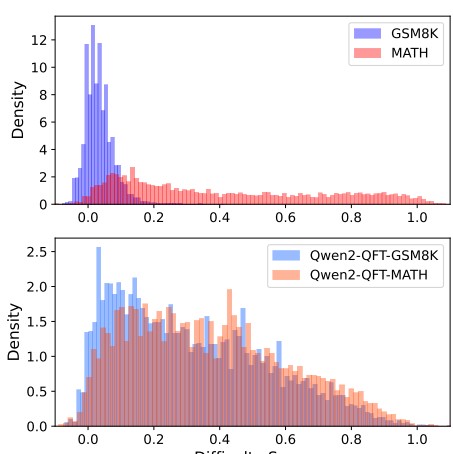

Figure 3: The difficulty distribution of two real-world datasets and two synthetic datasets. The difficulty score is calculated based solely on the problem part.

question distribution, we used a difficulty classifier, which maps a question into a difficulty score (details in Section 2.4). We performed QFT based on Qwen2-Math-7B (Yang et al., 2024a), then used the two QFT models, `Qwen2-QFT-GSM8K` and `Qwen2-QFT-MATH`, to synthesize 10K questions. The difficulty distribution of these four datasets is shown in Figure 3. We found that the generated questions separately differed from both GSM8K and MATH, yet they both converged toward

the same distribution. Additionally, the QFT model, trained on English questions, demonstrated the ability to generate a substantial number of questions in other languages. Both phenomena suggest that the QFT process enhances the model's question-generation capabilities without leading to overfitting the training data.

### 2.3 QUESTION PREFERENCE OPTIMIZATION (QPO)

The model is able to generate meaningful and diverse questions after QFT, but the quality is still not high enough, as shown in Figure 2. This is reflected in two aspects: (1) solvability: the math problem should have appropriate constraints and correct answers, and (2) difficulty: the model needs to learn from more challenging problems, yet some of the generated questions are still too simple. To address these two aspects, we applied Question Preference Optimization (QPO).

We first used the model after QFT to generate 10K questions. Then, we optimized these samples using an external LLM, focusing primarily on solvability and difficulty. We found that simultaneously optimizing both posed a challenge for the LLMs. Therefore, for each sample, we randomly selected one of the two optimization directions, prioritizing either solvability or difficulty. The optimization prompts can be found in Figure 10 and 11. The optimized questions, denoted as $y_w$, are treated as preferred

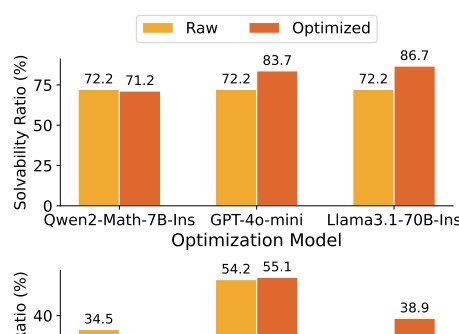

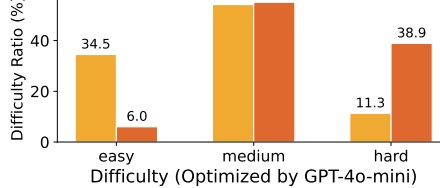

Figure 4: The solvability and difficulty of the raw questions generated by the QFT model and the optimized ones.

data, while the original questions before optimization, denoted as $y_l$, are considered dispreferred data. We modified the loss for Direct Preference Optimization (DPO) (Rafailov et al., 2024) formulation to fit our approach:

$$\mathcal{L}_{\text{QPO}}(\pi_\theta; \pi_{\text{ref}}) = -\mathbb{E}_{(y_w, y_l) \sim \mathcal{D}} \left[ \log \sigma \left( \beta \log \frac{\pi_\theta(y_w)}{\pi_{\text{ref}}(y_w)} - \beta \log \frac{\pi_\theta(y_l)}{\pi_{\text{ref}}(y_l)} \right) \right]. \tag{2}$$

The question optimization process placed significant demands on the model's ability to follow complex instructions. We experimented with three question optimization models: Qwen2-Math-7B-Instruct, GPT-4o-mini and Llama3.1-70B-Ins. To evaluate improvements in solvability and difficulty, we used GPT-4o, with the prompts for this evaluation provided in Figure 12 and 13. The results are shown in Figure 4. In terms of solvability, Qwen2-Math-7B-Instruct proved inadequate for this task, as the optimized questions resulted in decreased solvability. A possible reason for this is the model's insufficient ability to follow instructions accurately, resulting in many answers that fail to meet the specified optimization constraints. Considering the cheap API calls, we selected GPT-4o-mini as the question optimization model consequently.

### 2.4 QUESTION FILTERING

After the QFT and QPO phases, we obtained two question generators: DeepSeekMath-QGen and Qwen2-Math-QGen. There are still some minor issues in the generated questions, primarily related to language, solvability, and difficulty. To address these challenges, we applied the following filtering steps:

**Language Filtering** The question generator models still produce a substantial number of math questions in other languages, accounting for approximately 20%. Since our focus is on English math questions, we removed non-English questions by identifying questions containing non-English characters and filtering out those samples.

**Solvability Filtering** Although QPO effectively enhances the solvability of generated questions, some questions remain nonsensical. This is primarily due to (1) poorly constrained questions, where

missing conditions, redundant conditions, or logical inconsistencies occur, and (2) questions that do not yield meaningful outcomes (e.g., answers involving the number of people should result in a non-negative integer). To filter out such samples, we used Qwen2-Math-7B-Instruct to evaluate whether the question is meaningful and whether the conditions are sufficient. The prompts used for the solvability check are provided in Figure 12.

**Difficulty Sampling**  We measure the difficulty of a question using the fail rate (Tong et al., 2024) — the proportion of incorrect responses when sampling $n$ responses for a given question. This metric aligns with the intuition that harder questions tend to result in fewer correct responses. Following Tong et al. (2024), we used DeepseekMath-7B-RL as the sampling model to evaluate the difficulty of each question in the training sets of GSM8K and MATH, obtaining the fail rate for each question as its difficulty score. We then used this data to train a difficulty scorer. Specifically, we built upon DeepseekMath-7B-Base and added a classification head on top of the model's hidden state. The difficulty score $d$ is computed and optimized as:

$$d = Wh_l + b, \mathcal{L} = \frac{1}{N} \sum_{i=1}^{N} (y_i - d_i)^2,$$  (3)

where $W$ and $b$ are the weights and biases of the classification head, $h_l$ represents the last hidden state of the sequence, and $d_i$ is the predicted difficulty score for the $i$-th question. The loss function $\mathcal{L}$ is the mean squared error (MSE), where $y_i$ represents the true difficulty score for the $i$-th question. We then used the scorer to predict the difficulty of each synthetic question and sample based on the question's difficulty. Specifically, we filtered out a portion of the questions generated by DeepSeekMath-QGen that were overly simple. In contrast, the difficulty distribution of Qwen2-Math-QGen was more balanced, so no sampling was necessary.

### 2.5 RESPONSE GENERATION WITH REWARD FILTERING

Prior efforts to guarantee the quality of solutions include two aspects: (1) rejection sampling (Yuan et al., 2023): Large language models (LLMs) are tasked with generating multiple responses, specifically reasoning paths, for each instruction. Only reasoning paths that lead to the correct answer are preserved as solutions (Tong et al., 2024). (2) If the correct answer is unavailable, a majority voting method is used (Huang et al., 2024a), selecting the answer that appears most frequently across multiple reasoning paths and retaining these as the solutions. We use the reward model score as a metric for evaluating the quality of responses, considering its broader applicability, as there is often no single correct answer in other reasoning tasks like code generation and tool planning. Specifically, for each question, we generate 5 solutions and select the solution with the highest reward model scores as the preferred solution. In our experiments, we use InternLM2-7B-Reward (Cai et al., 2024) as our reward model. This choice was primarily guided by the model's performance on the reasoning subset of the Reward Bench (Lambert et al., 2024).

## 3 EXPERIMENT

### 3.1 EXPERIMENTAL SETUP

**Training Problem Designers**  Our question synthesis process relies on two problem designer models: `Deepseek-QGen` and `Qwen2-Math-QGen`, which were trained using QFT (section 2.2) and QPO (section 2.3), based on DeepSeekMath-7B-RL (Shao et al., 2024) and Qwen2-Math-7B-Instruct (Yang et al., 2024a), respectively. During the QFT stage, both models are trained on a mixed training subset of GSM8K and MATH problems, containing a total of 15K problems. We trained for only 1 epoch, considering that training for more epochs might cause the models to overfit the training problems and negatively impact the diversity of generated questions. We also used sequence packing (Krell et al., 2021) to accelerate training. In the QPO stage, we use 10K preference data for training, with a learning rate of 5e-7 and a batch size of 128.

**Question Generation**  The two question generation models were then utilized to generate a total of 2 million questions, with 1 million from each model. During this process, we set the maximum

Table 1: Main results on four mathematical reasoning benchmarks. **Bold** means the best score within the respective base model. The baselines use different synthesis models for both question synthesis and response generation, such as GPT-3.5, GPT-4, and GPT-4o. For our approach, DSMath-7B-QGen and Qwen2-Math-7B-QGen are utilized for question synthesis, with Qwen2-Math-7B-Ins used for response generation. If multiple models are used, only the latest released one is marked. More details concerning these datasets are shown in Figure 6.

| Model | Synthesis Model | GSM8K | MATH | College Math | Olympiad Bench | Average |
|---|---|---|---|---|---|---|
| *Teacher Models in Data Synthesis* | | | | | | |
| 🌀 GPT-4-0314 | - | 94.7 | 52.6 | 24.4 | - | - |
| 🌀 GPT-4-Turbo-24-04-09 | - | 94.5 | 73.4 | - | - | - |
| 🌀 GPT-4o-2024-08-06 | - | 92.9 | 81.1 | 50.2 | 43.3 | 66.9 |
| 🐋 DeepSeekMath-7B-RL | - | 88.2 | 52.4 | 41.4 | 19.0 | 49.3 |
| 🦅 Qwen2-Math-7B-Instruct | - | 89.5 | 73.1 | 50.5 | 37.8 | 62.7 |
| *General Base Model* | | | | | | |
| Mistral-7B-WizardMath | 🌀 GPT-4 | 81.9 | 33.3 | 21.5 | 8.6 | 36.3 |
| Mistral-7B-MetaMath | 🌀 GPT-3.5 | 77.7 | 28.2 | 19.1 | 5.8 | 32.7 |
| Mistral-7B-MMIQC | 🌀 GPT-4 | 75.7 | 36.3 | 24.8 | 10.8 | 36.9 |
| Mistral-7B-MathScale | 🌀 GPT-3.5 | 74.8 | 35.2 | 21.8 | - | - |
| Mistral-7B-KPMath | 🌀 GPT-4 | 82.1 | 46.8 | - | - | - |
| Mistral-7B-DART-Math | 🐋 DSMath-7B-RL | 81.1 | 45.5 | 29.4 | 14.7 | 42.7 |
| Mistral-7B-NuminaMath | 🌀 GPT-4o | 82.1 | 49.4 | 33.8 | 19.4 | 46.2 |
| Mistral-7B-ScaleQuest | 🦅 Qwen2-Math-7B-Ins | **88.5** | **62.9** | **43.5** | **26.8** | **55.4** |
| Llama3-8B-MetaMath | 🌀 GPT-3.5 | 77.3 | 32.5 | 20.6 | 5.5 | 34.0 |
| Llama3-8B-MMIQC | 🌀 GPT-4 | 77.6 | 39.5 | 29.5 | 9.6 | 39.1 |
| Llama3-8B-DART-Math | 🐋 DSMath-7B-RL | 81.1 | 46.6 | 28.8 | 14.5 | 42.8 |
| Llama3-8B-NuminaMath | 🌀 GPT-4o | 77.2 | 50.7 | 33.2 | 17.8 | 44.7 |
| Llama3-8B-ScaleQuest | 🦅 Qwen2-Math-7B-Ins | **87.9** | **64.4** | **42.8** | **25.3** | **55.1** |
| *Math-Specialized Base Model* | | | | | | |
| DeepSeekMath-7B-Instruct | - | 82.7 | 46.9 | 37.1 | 14.2 | 45.2 |
| DeepSeekMath-7B-MMIQC | 🌀 GPT-4 | 79.0 | 45.3 | 35.3 | 13.0 | 43.2 |
| DeepSeekMath-7B-KPMath-Plus | 🌀 GPT-4 | 83.9 | 48.8 | - | - | - |
| DeepSeekMath-7B-DART-Math | 🐋 DSMath-7B-RL | 86.8 | 53.6 | 40.7 | 21.7 | 50.7 |
| DeepSeekMath-7B-Numina-Math | 🌀 GPT-4o | 75.4 | 55.2 | 36.9 | 19.9 | 46.9 |
| DeepSeekMath-7B-ScaleQuest | 🦅 Qwen2-Math-7B-Ins | **89.5** | **66.6** | **47.7** | **29.9** | **58.4** |
| Qwen2-Math-7B-MetaMath | 🌀 GPT-3.5 | 83.9 | 49.5 | 39.9 | 17.9 | 47.8 |
| Qwen2-Math-7B-DART-Math | 🐋 DSMath-7B-RL | 88.6 | 58.8 | 45.4 | 23.1 | 54.0 |
| Qwen2-Math-7B-Numina-Math | 🌀 GPT-4o | 84.6 | 65.6 | 45.5 | 33.6 | 57.3 |
| Qwen2-Math-7B-ScaleQuest | 🦅 Qwen2-Math-7B-Ins | **89.7** | **73.4** | **50.0** | **38.5** | **62.9** |

generation length to 512, a temperature of 1.0, and a top-p value of 0.99. To ensure quality, we applied a question filtering pipeline (section 2.4) that involved language filtering, solvability filtering, and difficulty sampling. This process refined the dataset, leaving approximately 1M questions to form the final question pool, 400K from Deepseek-QGen and 600K from Qwen2-Math-QGen.

**Response Generation** Based on the problems, we synthesized responses (section 2.5) using Qwen2-Math-7B-Instruct (Yang et al., 2024a). In the process, we set the maximum generation length to 2048, with a temperature of 0.7 and top-p of 0.95. We use chain-of-thought prompt (Wei et al., 2022) to synthesize solutions. We use vLLM (Kwon et al., 2023) to accelerate the generation and Ray (Moritz et al., 2018) to deploy distributed inference. For each problem, we sampled 5 solutions and selected the one with the highest reward score as the final response. The final dataset consists of 1 million problem-solution pairs.

**Instruction Tuning** We conducted instruction tuning on the synthetic problems and solutions using two general base models, Mistral-7B (Jiang et al., 2023) and Llama3-8B (Dubey et al., 2024), as well as two math-specialized base models, DeepSeekMath-7B (Shao et al., 2024) and Qwen2-Math-7B (Yang et al., 2024a). All models were fine-tuned for 3 epochs in our experiments unless specified otherwise. We used a linear learning rate schedule with a 3% warm-up ratio, reaching a peak of 5e-5 for Llama3 and DeepSeekMath and 1e-5 for the other models, followed by cosine decay to zero.

**Evaluation and Metrics**   We assessed the fine-tuned models' performance across four datasets of increasing difficulty. Along with the widely used GSM8K (elementary level) and MATH (competition level), we included two more challenging benchmarks: College Math (Yuan et al., 2023) (college level) and Olympiad Bench (He et al., 2024) (Olympiad level). For evaluation, we employed the script from Tong et al. (2024) to extract final answers and determine correctness by comparing answer equivalency. The generated outputs were all in the form of natural language Chain-of-Thought (CoT) reasoning (Wei et al., 2022) through greedy decoding, with no tool integration, and we report zero-shot pass@1 accuracy.

**Compared Baselines**   The main point of comparison is data synthesis methods, including: (1) WizardMath (Luo et al., 2023) proposes a reinforced Evol Instruct method; (2) MetaMath (Yu et al., 2023a) introduces three types of question bootstrapping; (3) MMIQC (Liu & Yao, 2024) proposes an iterative question composing method; (4) Orca-Math (Mitra et al., 2024) augments existing datasets using an Agent-Instruct method; (5) KPMath (Huang et al., 2024a) utilizes inherent topics and key points to synthesize problems; and (6) MathScale (Tang et al., 2024) builds a concept graph to generate new questions. In addition to this, we also involved other large math corpus like (7) DART-Math (Tong et al., 2024) enhances the response generation process through difficulty-guided rejection sampling; (8) Numina-Math (Li et al., 2024c) collects a large corpus by combining existing synthetic data with real-world datasets. More details of these datasets are shown in Table 6. We found that different scripts yielded varying evaluation results. To ensure consistency, we evaluated all released models using the same evaluation scripts. For methods without available results or released models, we retrained the models using their publicly available data.

## 3.2   MAIN RESULTS

**ScaleQuest significantly outperforms others**   Table 1 presents the results. ScaleQuest significantly outperforms previous synthetic methods, with average performance improvements ranging from 5.6% to 11.5% over the prior state-of-the-art (SoTA) on both general base models and math-specialized foundation models. Qwen2-Math-7B-ScaleQuest achieved a zero-shot pass@1 accuracy of 73.4 on the MATH benchmark, matching the performance of GPT-4-Turbo. For out-of-domain tasks, Qwen2-Math-7B-ScaleQuest outperformed its teacher model, Qwen2-Math-7B-Instruct, with scores of 89.7 on the GSM8K benchmark, 73.4 on the MATH benchmark, and 38.5 on the Olympiad benchmark. It's important to highlight that Qwen2-Math-7B-Instruct has undergone Group Relative Policy Optimization (GRPO) (Shao et al., 2024), utilizing the powerful reward model Qwen2-Math-RM-72B (Yang et al., 2024a), while our model is only an instruction tuning version. To ensure a fair comparison with other baselines, we have only applied supervised fine-tuning (SFT) in this work, leaving the preference tuning process for future work. Since some of the baseline datasets are not publicly available, we could not strictly control for the same training data volume. Experiments with controlled training data volumes can be found in the scaling trend in Figure 1 and Appendix C. The use of multiple models in our approach may lead to some confusion. To address this, we also present a simplified version of our method, where only Qwen2-Math-7B-Ins and InternLM-7B-Reward are used. Detailed results and more insights into the selection of these models are provided in Appendix C.

**ScaleQuest scales well with increasing data**   We also explored the scalability of our dataset. We used our constructed dataset along with publicly available datasets, including MetaMath (Yu et al., 2023a), DART-Math (Tong et al., 2024), and Numina-Math (Li et al., 2024c). We trained the model using Llama3-8B and observed how its performance scaled with increasing data size. The results are presented in Figure 1. For the in-domain evaluation (MATH), our method demonstrates high data efficiency, achieving superior results with the same amount of data. In out-of-domain evaluations (Olympiad Bench), it also shows strong scalability, continuing to improve even as other datasets reach their limits. A limited question set leads to constrained improvements in model performance, as demonstrated by the results of DART-Math, which relies on a small number of questions and generates numerous correct answers through rejection sampling. Limited questions face a scalability ceiling, as the lack of diversity in the question set restricts further performance growth. Our results further demonstrate that diverse questions support sustained performance growth, emphasizing the need for broader and more varied question generation.

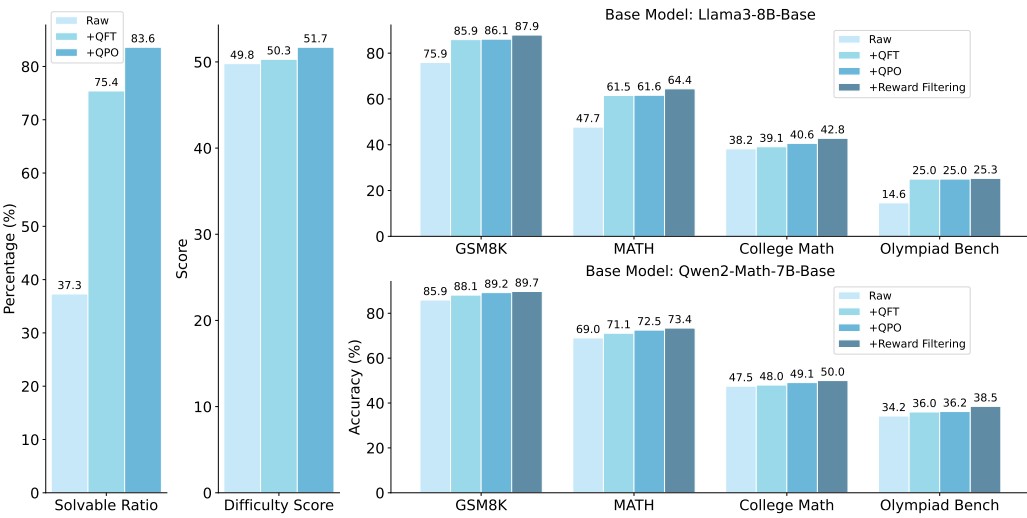

Figure 5: A comparison of the synthetic dataset generated by the raw instruct model, the model after QFT, the model after QPO, and the final dataset after applying reward filtering. **Left:** The solvable ratio and difficulty score of the generated questions. The solvable ratio refers to the proportion of generated questions that are judged as "solvable", while the difficulty score represents the average difficulty rating assigned to each generated question. **Right:** The instruction tuning effectiveness on Llama3-8B and Qwen2-Math-7B.

Table 2: We directly compared the question quality of different open-source datasets. To ensure consistency, all responses were generated using Qwen2-Math-7B-Instruct with the same reward filtering process.

| Questions Source | Response Synthesis Model | GSM8K | MATH | College Math | Olympiad Bench | **Average** |
|---|---|---|---|---|---|---|
| MetaMath | 🦅 Qwen2-Math-7B-Instruct | 84.5 | 53.8 | 40.1 | 22.1 | 50.1 |
| OrcaMath | 🦅 Qwen2-Math-7B-Instruct | 84.2 | 53.7 | 40.5 | 23.7 | 50.5 |
| NuminaMath | 🦅 Qwen2-Math-7B-Instruct | 86.0 | 65.9 | 46.1 | **30.2** | 57.1 |
| ScaleQuest | 🦅 Qwen2-Math-7B-Instruct | **89.5** | **66.6** | **47.7** | 29.9 | **58.4** |

### 3.3 ABLATION STUDY

**Ablation on each sub-method** To validate the effectiveness of each of our sub-methods, including QFT, QPO, and reward filtering, we conducted an ablation study. We evaluated the quality of the questions generated by the models across three dimensions: solvability, difficulty, and performance in instruction tuning. To assess the model's solvability and difficulty, we used GPT-4o-mini as the evaluation model, with the prompts provided in the Figure 12 and 13. For difficulty evaluation, we calculated the dataset's average difficulty score based on ratings for each question: "very easy" is rated as 20 points, "easy" as 40 points, "medium" as 60 points, "hard" as 80 points, and "very hard" as 100 points.

The results are shown in Figure 5. The "raw model" refers to using the instruct model to directly generate instructions and responses, as done in Xu et al. (2024). To ensure fairness, we also generated 1M question-response pairs using their method based on Qwen2-Math-7B-Instruct, which were used to train Llama3-8B. After applying QFT and QPO, the model's performance improved across all three evaluation dimensions, demonstrating the effectiveness of our approach. Furthermore, by filtering for solvable questions and applying reward filtering to the responses, the quality of our dataset increased, resulting in significant improvements across all four evaluation benchmarks.

**Question matters for data synthesis** To directly compare the question quality of our constructed data with other open-source datasets, we used the same model, Qwen2-Math-7B-Instruct, to gener-

Table 3: The performance of Mistral-7B-v0.1 fine-tuned on ScaleQuest-DSMath, ScaleQuest-Qwen2, and a mix of both. In this setup, the instructions for ScaleQuest-DSMath and ScaleQuest-Qwen2-Math were generated by DSMath-QGen and Qwen2-Math-QGen, respectively. We fixed the training data size at 400K and found that the mixed data resulted in the greatest improvement.

| Synthetic Dataset | # Samples | GSM8K | MATH | College Math | Olympiad Bench | **Average** |
|---|---|---|---|---|---|---|
| ScaleQuest-DSMath | 400K | 87.6 | 52.2 | 39.8 | 19.4 | 49.8 |
| ScaleQuest-Qwen2-Math | 400K | 86.8 | 56.1 | 39.6 | 18.7 | 50.3 |
| Mixed | 400K | **87.8** | **58.0** | **40.1** | **22.2** | **52.0** |

Table 4: Cost analysis of the entire data synthesis process. We also estimated the cost of generating the same number of tokens using proprietary models GPT-4 and GPT-4o for comparison.

| Phase | | Type | # Samples | GPU hours | Cost ($) |
|---|---|---|---|---|---|
| QFT | Training DSMath-QFT | Train | 15K | 2.0 | 2.6 |
| | Training Qwen2-Math-QFT | Train | 15K | 1.9 | 2.5 |
| QPO | Generate Questions | Infer | 10K×2 | 0.4 | 0.5 |
| | Construct Preference Data | API | 10K×2 | - | 6.2 |
| | QPO Training | Train | 10K×2 | 6.6 | 8.5 |
| Data Synthesis | Question Generation | Infer | 2M | 38.4 | 49.5 |
| | solvability & difficulty check | Infer | 2M | 110.6 | 142.7 |
| | Response Generation | Infer | 1M×5 | 251.0 | 323.8 |
| | Reward Scoring | Infer | 1M×5 | 112.0 | 144.5 |
| **Total** | | | 1M | 522.9 | 680.8 |
| GPT-4 cost (generating the same number of tokens) | | | - | - | 24,939.5 |
| GPT-4o cost (generating the same number of tokens) | | | - | - | 6,115.9 |

ate responses and fine-tuned DeepSeekMath-7B based on the synthetic datasets. As shown in Table 2, using the same response generation method, our model outperformed other synthetic datasets like MetaMath and OrcaMath, highlighting the high quality of our questions. NuminaMath also demonstrated competitive performance, largely due to the fact that many of its questions are drawn from real-world scenarios. This also highlights that question quality is crucial for synthetic data.

**Multiple question generators enhance data diversity**   We use two models as question generators: DSMath-QGen and Qwen2-Math-QGen, which are based on DeepSeekMath (Shao et al., 2024) and Qwen2-Math (Yang et al., 2024a), respectively. To explore the impact of using multiple question generators, we compared the effects of using data synthesized by a single generator versus a mix of data from both. We fixed the total dataset size at 400K and used it to fine-tune Mistral-7B. As shown in Table 3, we found that the mixed data outperformed the data generated by either single generator. A possible explanation for this improvement is the increased data diversity. In fact, we observed that DSMath-QGen tends to generate simpler, more real-world-oriented questions, while Qwen2-Math-QGen produces more challenging, theory-driven ones. From this, we recognize the potential of using multiple question generators, and we plan to incorporate more question generators as part of our future work.

## 3.4   COST ANALYSIS

The data synthesis process was conducted on a server with 8 A100-40G-PCIe GPUs. We summarize our overall costs in Table 4. Generating 1 million data samples required only 522.9 GPU hours (approximately 2.7 days on an 8-GPU server), with an estimated cost of $680.8 for cloud server rental.[2]

---

[2] https://lambdalabs.com/service/gpu-cloud

This is only about 10% of the cost of generating the same data using GPT-4o. This demonstrates that our data generation method is significantly more cost-effective.

# 4 RELATED WORK

## 4.1 MATHEMATICAL REASONING

Solving math problems is regarded as a key measure of evaluating the reasoning ability of LLMs. Recent advancements in mathematical reasoning for LLMs, including models like OpenAI o1, Claude-3.5, Gemini (Reid et al., 2024), DeepSeekMath (Shao et al., 2024), InternLM2-Math (Cai et al., 2024), and Qwen2.5-Math (Yang et al., 2024b), have spurred the development of various approaches to improve reasoning capabilities of LLMs on math-related tasks. To strengthen the math reasoning capabilities of LLMs, researchers have focused on areas such as prompting techniques (Chia et al., 2023; Chen et al., 2023; Zhang et al., 2023), data construction for pretraining (Lewkowycz et al., 2022; Azerbayev et al., 2023; Zhou et al., 2024; Shao et al., 2024) and instruction tuning (Luo et al., 2023; Yue et al., 2023), tool-integrated reasoning(Chen et al., 2022; Gao et al., 2023; Gou et al., 2023; Wang et al., 2023; Yue et al., 2024; Yin et al., 2024; Zhang et al., 2024), and preference tuning (Ma et al., 2023; Luong et al., 2024; Shao et al., 2024; Lai et al., 2024). Our work primarily focuses on math data synthesis for instruction tuning.

## 4.2 DATA SYNTHESIS FOR MATH INSTRUCTION TUNING

High-quality reasoning data, particularly well-crafted questions, is in short supply. Prior efforts have mostly started with a small set of human-annotated seed instructions and expanded them through few-shot prompting. We categorize them into two types: question-driven augmentation and knowledge-driven augmentation. Previous works focus on enhancing seed questions by introducing additional constraints or numerical changes to increase the reasoning steps required. For instance, WizardMath (Luo et al., 2023) uses a series of operations to increase the complexity of questions and answers with GPT-3.5. MetaMath (Yu et al., 2023a) enhances the questions in GSM8K (Cobbe et al., 2021) and MATH (Hendrycks et al., 2021) by rewriting them in various ways, such as through semantic rephrasing, self-verification, and backward reasoning. Xwin-Math (Li et al., 2024a) and MMIQC (Liu & Yao, 2024) further explore the scalability of the synthetic data. However, these methods face a diversity challenge, as few-shot prompting often results in new instructions that are too similar to the original seed questions (Li et al., 2024b). To increase diversity, recent works have focused on knowledge-driven data synthesis, where they summarize world knowledge from the seed questions and use it to generate synthetic datasets (Didolkar et al., 2024; Shah et al., 2024). MathScale (Tang et al., 2024) extracts math concepts from seed questions and then generate math reasoning data. KPMath (Huang et al., 2024a) begins by extracting topics and key points from seed problems using a labeling model, and sample multiple topics and key points for instruction synthesis. There are other methods for enhancing dataset quality as well. DART-Math (Tong et al., 2024) focuses on enhancing the quality of responses by using rejection sampling to generate multiple correct answers for each query from GSM8K and MATH. In contrast, Numina-Math (Li et al., 2024c) improves its dataset by collecting more real-world and synthetic data, then reformatting (Fan et al., 2024) the responses using GPT-4o. This high-quality data can be integrated with our constructed dataset, resulting in an improved data mix for more effective instruction tuning.

# 5 CONCLUSION

In this work, we propose ScaleQuest, a novel data synthesis framework that unlocks the ability of open-source smaller models to independently generate large-scale, high-quality reasoning data from scratch, at a low cost. By training the problem-solving models on a small subset of questions, we effectively activate their question-generation capabilities. We also introduce a response enhancement method. With these techniques, we successfully developed a fully synthetic math reasoning dataset consisting of 1 million question-answer pairs. Using this dataset, we fine-tuned the model and achieved remarkable improvements, with gains ranging from 29.2% to 46.4% compared to the base model. The fine-tuned 7B model, Qwen2-Math-7B-ScaleQuest, outperforms all competitors in the 7B-70B range and even surpasses proprietary models like GPT-4-Turbo and Claude-3.5-Sonnet.

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

## A    ADDITIONAL DATA STATISTICS

**Filtering process**    The entire data generation process is illustrated in Figure 6. After using the two question generators to produce 2 million questions from scratch, we performed a filtering process, including language filtering, solvability checks, and difficulty sampling. These steps filtered out 20.1%, 19.4%, and 9.2% of the samples, respectively, resulting in a final question set of 1 million questions. In the subsequent response generation process, we filtered out responses without answers by checking for key phrases such as "The answer is" or "\boxed{}". This step eliminated a negligible portion of the samples, as most of the filtered questions were solvable and did not pose any confusion for the response generation model.

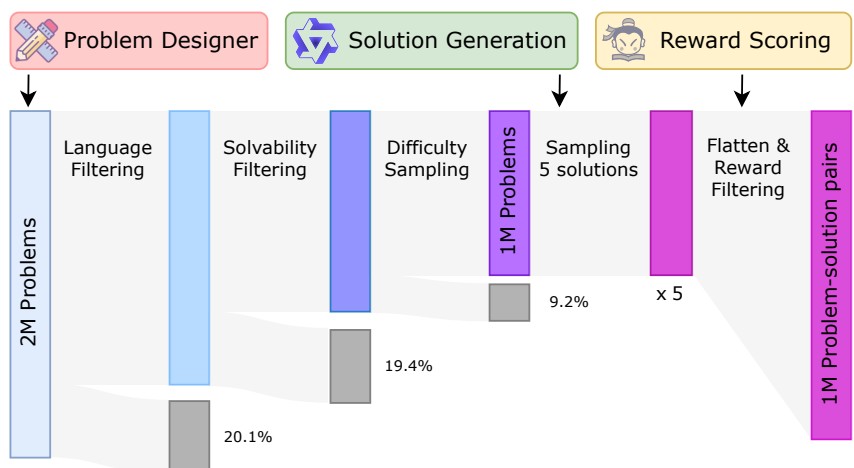

Figure 6: Overview of our filtering process.

**Dataset Coverage**    We analyze the dataset coverage through two aspects: (1) Problem Topic Coverage, such as algebra and geometry. Following Huang et al. (2024a), we use GPT-4o to categorize the topics of the given questions, with prompt illustrated in Figure 14. Figure 7 presents the results. We found that the topics covered the major areas of mathematics, such as arithmetic, algebra, geometry, and others. (2) Embedding space analysis. Following Zhao et al. (2024) and Xu et al. (2024), we first compute the input embeddings of the questions and then project them into a two-dimensional space using t-SNE (Van der Maaten & Hinton, 2008). We included only real-world datasets, such as GSM8K (Cobbe et al., 2021), MATH (Hendrycks et al., 2021), and NuminaMath (Li et al., 2024c) (which contains a small portion of synthetic questions). As shown in Figure 8, our synthetic data closely resembles the real-world questions.

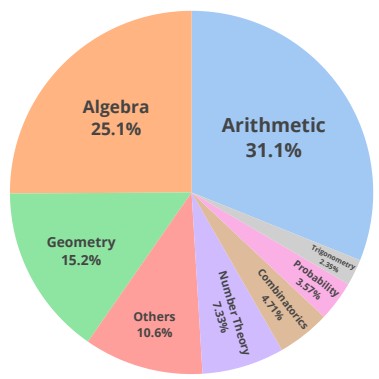

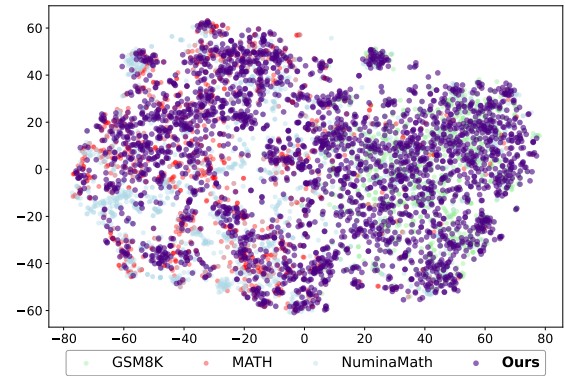

Figure 7: Topic distribution of our generated dataset.

Figure 8: t-SNE plot of our dataset, with GSM8K, MATH, and NuminaMath.

**Data Leakage Analysis**    We conducted an n-gram similarity analysis between the generated questions and all test sets from both our dataset and other baseline datasets. Based on prior empirical analysis (Brown, 2020; Wei et al., 2021), we set n=13 to prevent spurious collisions and calculated how much the test sets overlap with training data to assess data contamination. Table 5 shows the clean ratio of our dataset and other baseline datasets. The results demonstrate that our dataset achieves a relatively high level of data cleanliness compared to other datasets, suggesting that our method generates novel questions instead of memorizing existing ones.

Table 5: Overlap statistics for the datasets used. We report the clean ratio of the test set, representing the percentage of test samples that have no matching n-grams with samples in the training set.

| Dataset | GSM8K | MATH | College Math | Olympiad Bench | Average |
|---|---|---|---|---|---|
| MetaMath | 99.8 | 92.2 | 100 | 99.7 | 97.9 |
| NuminaMath | 99.8 | 89.8 | 99.9 | 86.8 | 94.1 |
| DART-Math | 99.8 | 91.5 | 100.0 | 99.6 | 97.7 |
| MMIQC | 99.8 | 88.0 | 98.9 | 97.9 | 96.2 |
| ScaleQuest (Ours) | 99.9 | 92.8 | 99.8 | 97.2 | 97.4 |

**Safety Analysis**    We used Llama3-8B-Guard (Inan et al., 2023) as a discriminator model to detect any unsafe elements in the data. After sampling 10K instances from the 1 million samples, we found that only 0.1% were flagged as unsafe.

**Generated Examples**    We sampled several generated examples from our datasets, as shown in Figure 17, 18 and 19. The generated math problems are of high quality, driving effective learning.

Table 6: Comparison between our constructed dataset and previous datasets.

| Dataset | Size | Synthesis Model | Public |
|---|---|---|---|
| WizardMath (Luo et al., 2023) | 96K | GPT-4 | ✗ |
| MetaMath (Yu et al., 2023a) | 395K | GPT-3.5-Turbo | ✓ |
| MMIQC (Liu & Yao, 2024) | 2294K | GPT-4 & GPT-3.5-Turbo & Human | ✓ |
| Orca-Math (Mitra et al., 2024) | 200K | GPT-4-Turbo | ✓ |
| Xwin-Math (Li et al., 2024a) | 1440K | GPT-4-Turbo | ✗ |
| KPMath-Plus (Huang et al., 2024a) | 1576K | GPT-4 | ✗ |
| MathsScale (Tang et al., 2024) | 2021K | GPT-3.5 & Human | ✗ |
| DART-Math (Tong et al., 2024) | 585K | DeepSeekMath-7B-RL | ✓ |
| Numina-Math (Li et al., 2024c) | 860K | GPT-4 & GPT-4o | ✓ |
| ScaleQuest | 1000K | DeepSeekMath-7B-RL Qwen2-Math-7B-Instruct | ✓ |

# B    DATA SYNTHESIS FOR CODE REASONING TASK

We also extend our ScaleQuest method to the Code Reasoning Task as a simple validation. We made the following modifications to adapt to the code reasoning task:

**Settings**    We choose DeepSeek-Coder-7B-Instruct (Guo et al., 2024) and Qwen2.5-Coder-7B-Instruct (Hui et al., 2024) as two problem-solving models to perform question fine-tuning on 20K questions randomly sampled from CodeFeedBack (Zheng et al., 2024). For Question Preference Optimization, we also focused on solvability and difficulty, making slight modifications to the prompts based on the code reasoning task. Our evaluation covered HumanEval (Chen et al., 2021), MBPP (Austin et al., 2021), and BigCodeBench (Zhuo et al., 2024), using the same evaluation script as Qwen2.5-Coder. We report pass@1 results using greedy search.

The results are presented in Table 7. Compared to the widely used refined version of CodeFeedback, namely CodeFeedback-Filtered, our generated data outperforms it, with an average improvement of 5.9 across the three baselines. Additionally, we enhanced the Response portion of CodeFeedback-Filtered using Qwen2.5-Coder-7B-Instruct, and the results indicate that our generated questions are of higher quality. This further demonstrates the effectiveness of the ScaleQuest method.

Table 7: Results of ScaleQuest in Code Reasoning Task. All results are based on Qwen2.5-Coder-7B-Base. CFB refers to the CodeFeedBack-Filtered Dataset. we augmented the responses for the problems in CodeFeedback-Filtered using Qwen2.5-Coder-7B-Instruct with reward filtering, creating a new dataset referred to as CFB-Aug.

| Model | # Samples (K) | HumanEval | MBPP | BigCodeBench | Average |
|---|---|---|---|---|---|
| Qwen2.5-Coder-CFB | 156 | 79.3 | 77.2 | 35.6 | 64.0 |
| Qwen2.5-Coder-CFB-Aug | 156 | 84.1 | **84.7** | 39.0 | 69.3 |
| Qwen2.5-Coder-ScaleQuest | 156 | **86.6** | 83.1 | **40.0** | **69.9** |

## C  MORE COMPARISON RESULTS

**Additional Results on Out-of-Domain (OOD) Benchmarks**  In addition to College Math and Olympiad Bench, we included two additional benchmarks: GSM-Hard (Gao et al., 2023) and Math-Chat (Liang et al., 2024). GSM-Hard is constructed by modifying the questions in GSM8K, replacing the numbers with larger, less common ones. From MathChat, we selected two problem-solving tasks: follow-up QA and error correction. The results are summarized in Table 8. In more fine-grained OOD evaluations, our model continues to perform on par with Qwen2-Math-7B-Ins, further demonstrating our ScaleQuest Model's generalization capability and highlighting the generated data's robustness.

Table 8: The comparison between Qwen2-Math-7B-Ins and the ScaleQuest Model on GSM-Hard and MathChat. We choose Follow-up QA and Error Correction from MathChat for evaluation in problem-solving. R1, R2, and R3 represent different rounds in Follow-up QA.

| Model | GSM-Hard | Follow-up QA | | | Error Correction | Average |
|---|---|---|---|---|---|---|
| | | R1 | R2 | R3 | | |
| Qwen2-Math-7B-Instruct | 68.3 | 89.5 | 62.4 | 53.5 | 89.9 | 72.7 |
| Qwen2-Math-7B-ScaleQuest | 66.3 | 89.7 | 61.7 | 53.5 | 91.1 | 72.5 |

**Comparison Under Equal Training Data Volume**  In the right panel of Figure 1, we plotted the scaling trends of model performance with increasing data volume, showcasing the superiority of the ScaleQuest method when using the same amount of data. To further ensure a fair comparison, we randomly sampled the same number of training examples from open-source datasets for training. Specifically, we sampled 400K examples from MetaMath, DART-Math, NuminaMath, and our dataset (for MetaMath, which contains 395K examples in total, all samples were used). The results are presented in Table 9. We observe that with the same amount of training data, our dataset demonstrates significantly higher instruction tuning effectiveness compared to other datasets.

**Insights behind model selection**  In our works, we use many models, e.g., DSMath-7B-RL, Qwen2-Math-7B-Ins, GPT-4o-mini, and DSMath-7B-Base, which may cause confusion for model selection. In response, we also supplemented our approach with a simpler setup. We used Qwen2-Math-7B-Ins for training question generators, constructing optimization data for QPO, and performing solvability & difficulty filtering, as well as for response generation. For reward filtering,

Table 9: Results on four mathematical reasoning benchmarks. All results are based on Qwen2-Math-7B-Base. ScaleQuest-Simple is a simplified version that only utilizes Qwen2-Math-7B-Ins for QFT, QPO, and question filtering, and InternLM-7B-Reward for reward filtering.

| Model | # Samples (K) | GSM8K | MATH | College Math | Olympiad Bench | Average |
|---|---|---|---|---|---|---|
| Qwen2-Math-7B-MetaMath | 395 | 84.3 | 48.6 | 40.5 | 15.6 | 47.3 |
| Qwen2-Math-7B-DART-Math | 400 | 88.6 | 58.2 | 45.2 | 22.8 | 53.7 |
| Qwen2-Math-7B-NuminaMath | 400 | 82.0 | 65.8 | 44.9 | 29.2 | 55.5 |
| Qwen2-Math-7B-ScaleQuest | 400 | **90.6** | **71.6** | **50.2** | **36.2** | **62.1** |
| Qwen2-Math-7B-ScaleQuest-Simple | 400 | 89.4 | 69.9 | 48.8 | 33.6 | 60.4 |

InternLM-7B-Reward remained unchanged. The results, as shown in Figure 9 (ScaleQuest-Simple result), indicate that our approach continues to demonstrate superior performance compared to existing datasets. Additionally, we summarize these insights on model selection for domain adaptation:

- Selection of base model for training question generator: The self-synthesis generation paradigm heavily relies on the inherent knowledge of the problem-solving model itself (Xu et al., 2024). Therefore, a domain-specific model is essential. For example, Qwen2-Math-Ins is suitable for mathematical reasoning, while Qwen2.5-Coder-Ins fits well for code reasoning. Furthermore, using multiple question generators often leads to more diverse and higher-quality questions (as discussed in section 3.3).

- Selection of model for constructing optimization data: Well-aligned, general-purpose models, such as Llama3.1-70B and GPT-4o-mini, tend to perform better than domain-specific models, as illustrated in Figure 4.

- Selection of Response Generation Model & Reward Model: These can be selected based on their performance on the corresponding mathematical tasks.

We believe that the methodology and the experience in selecting models are always more critical than the chosen models themselves. With the continuous advancements in the open-source community, we are confident that stronger models will undoubtedly produce even better datasets when applying our approach.

**More ablation Results of each submethod** In Figure 5, we discussed the effectiveness of each submethod, including Question Fine-Tuning (QFT), Question Preference Optimization (QPO), and Reward Filtering (RF), in a stepwise manner. To further refine this ablation study, we examined various combinations of these submethods. We excluded the combination of w/o QFT and w/ QPO, as QPO is meaningless without QFT, which is essential for question generation. The results are illustrated in Table 10. From the results, we can more precisely observe the contributions of each submethod to overall performance improvements. We found that QFT and QPO contribute significantly to the improvement of SFT performance, while the impact of QPO seems less pronounced. We would like to clarify that the limited improvements from QPO are due to two main reasons: (1) QPO primarily optimizes the solvability of questions, and its influence on response quality is indirect. (2) Though the impact of QPO in SFT may be minimal, it significantly enhances the data generation efficiency. Specifically, QPO improves the solvability of generated questions from 75.4% to 83.6%, a meaningful enhancement that boosts the efficiency of data utilization. While the effect may appear minimal due to subsequent solvability filtering, our detailed analysis shows that 28.8% of unsolvable questions were filtered out in the baseline setting, whereas after QPO, only 19.4% were deemed unsolvable. This represents a 9.4% reduction in computational overhead.

**Human Evaluation Results** We conducted a human evaluation of the generated data, focusing on three aspects: clarity, reasonableness, and real-world relevance. For reference, we also included two high-quality, human-curated datasets, GSM8K and MATH. A total of 40 examples were sampled from each dataset and evaluated based on clarity, coherence, and real-world relevance, with scores

Table 10: Results of various combinations of these submethods on MATH. All results are based on Llama3-8B.

| QFT | QPO | RF | GSM8K | MATH | College Math | Olympiad Bench | Average |
|-----|-----|-----|-------|------|--------------|----------------|---------|
| ✗ | ✗ | ✗ | 74.2 | 44.5 | 36.9 | 13.0 | 42.2 |
| ✗ | ✗ | ✓ | 75.9 | 47.7 | 38.2 | 14.6 | 44.1 |
| ✓ | ✗ | ✗ | 85.9 | 61.5 | 39.1 | 25.0 | 52.9 |
| ✓ | ✓ | ✗ | 86.1 | 61.6 | 40.6 | 25.0 | 53.3 |
| ✓ | ✗ | ✓ | 88.0 | 63.4 | 41.9 | 25.4 | 54.7 |
| ✓ | ✓ | ✓ | 87.9 | 64.4 | 42.8 | 25.3 | 55.1 |

ranging from 1 to 5. The results are presented in Table 11. In terms of clarity and reasonableness, our synthetic data surpasses NuminaMath but still falls short of the high-quality, real-world datasets like the training sets of GSM8K and MATH. Regarding real-world relevance, GSM8K leans toward practical, real-life scenarios, while MATH focuses more on theoretical mathematical derivations. Our generated data can be seen as a balance between the two.

Table 11: Human Evaluation Results.

| Dataset | clarity | reasonableness | real-world relevance |
|---------|---------|----------------|----------------------|
| GSM8K | 4.4 | 4.5 | 3.9 |
| MATH | 4.1 | 4.3 | 2.4 |
| NuminaMath | 3.8 | 4.0 | 2.4 |
| ScaleQuest | 3.9 | 4.0 | 2.8 |

**Effect of Training Data Volume on QPO**   QPO is designed to enhance the solvability and difficulty of the question generator. We investigate the impact of training data volume by using GPT-4o-mini as the optimization model. The training data volume was controlled at 5K, 10K, 15K, 20K, and 40K, with Qwen2-Math-7B-QFT serving as the base model. We evaluated the performance of the trained question generator in terms of solvability and difficulty. The results are shown in Figure 9. As the amount of training data increases, both the solvable rate and difficulty of the questions generated by the question generator improve, gradually converging around 20K training examples. We believe that maintaining the training data at approximately 10K represents a more suitable balance between training cost and model performance.

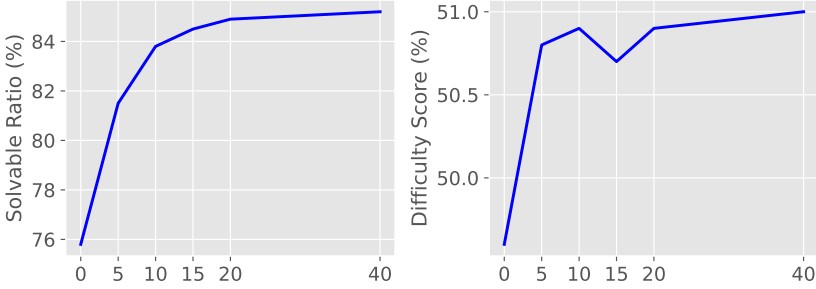

Figure 9: Performance of QPO in different training data volume. The evaluation covers the solvable ratio and difficulty score, following the same evaluation procedure as in Figure 5.

**Additional Results Based on Different Base Models**   We have supplemented Table 2 with the results for the other three base models, as shown in Table 12. Under the same response generation

Table 12: Additional results of Table 2 on the other base models. All responses were generated using Qwen2-Math-7B-Instruct with the same reward filtering process. For baseline datasets, "-Aug" indicates that the responses have been enhanced.

| Model | GSM8K | MATH | College Math | Olympiad Bench | Average |
|---|---|---|---|---|---|
| Mistral-7B-MetaMath-Aug | 77.0 | 34.1 | 18.6 | 8.6 | 34.6 |
| Mistral-7B-OrcaMath-Aug | 84.4 | 31.6 | 20.9 | 8.2 | 36.3 |
| Mistral-7B-NumiMath-Aug | 79.5 | 62.8 | 40.4 | **30.4** | 53.3 |
| Mistral-7B-ScaleQuest | **88.5** | **62.9** | **43.5** | 28.8 | **55.9** |
| Llama3-8B-MetaMath-Aug | 77.6 | 33.1 | 20.6 | 9.2 | 35.1 |
| Llama3-8B-OrcaMath-Aug | 83.2 | 32.6 | 19.4 | 8.6 | 36.0 |
| Llama3-8B-NumiMath-Aug | 79.1 | 62.9 | 39.3 | **25.4** | 51.7 |
| Llama3-8B-ScaleQuest | **87.9** | **64.4** | **42.8** | 25.3 | **55.1** |
| Qwen2-Math-7B-MetaMath-Aug | 88.5 | 68.5 | 47.1 | 33.0 | 59.3 |
| Qwen2-Math-7B-OrcaMath-Aug | 89.3 | 68.3 | 46.6 | 31.9 | 59.0 |
| Qwen2-Math-7B-NumiMath-Aug | 89.5 | 72.6 | 49.5 | 36.3 | 62.0 |
| Qwen2-Math-7B-ScaleQuest | **89.7** | **73.4** | **50.0** | **38.5** | **62.9** |

process, our approach consistently outperforms existing datasets across all four base models, further demonstrating the superiority of our method.

## D PROMPTS

---

**Prompts for Problem Solvability Optimization**

Please act as a professional math teacher.
Your goal is to create high quality math word problems to help students learn math.
You will be given a math question. Please optimize the Given Question and follow the instructions.
To achieve the goal, please follow the steps:
# Please check that the given question is a math question and write detailed solution to the Given Question.
# Based on the problem-solving process, double check the question is solvable.
# If you feel that the given question is not a meaningful math question, rewrite one that makes sense to you. Otherwise, modify the Given question according to your checking comment to ensure it is solvable and of high quality.
# If the question can be solved with just a few simple thinking processes, you can rewrite it to explicitly request multiple-step reasoning.

You have five principles to do this:
# Ensure the optimized question only asks for one thing, be reasonable and solvable, be based on the Given Question (if possible), and can be answered with only a number (float or integer). For example, DO NOT ask, 'what is the amount of A, B and C?'.
# Ensure the optimized question is in line with common sense of life. For example, the amount someone has or pays must be a positive number, and the number of people must be an integer.
# Ensure your student can answer the optimized question without the given question. If you want to use some numbers, conditions or background in the given question, please restate them to ensure no information is omitted in your optimized question.
# Please DO NOT include solution in your question.

Given Question: `problem`
Your output should be in the following format:
CREATED QUESTION: [your created question]
VERIFICATION AND MODIFICATION: [solve the question step-by-step and modify it to follow all principles]
FINAL QUESTION: [your final created question]

---

Figure 10: The prompts used to optimize the solvability of questions for QPO Training.

---

**Prompts for Problem Difficulty Optimization**

You are an Math Problem Rewriter that rewrites the given #Problem# into a more complex version.
Please follow the steps below to rewrite the given "#Problem#" into a more complex version.

Step 1: Please read the "#Problem#" carefully and list all the possible methods to make this problem more complex (to make it a bit harder for well-known AI assistants such as ChatGPT and GPT4 to handle). Note that the problem itself might be erroneous, and you need to first correct the errors within it.
Step 2: Please create a comprehensive plan based on the #Methods List# generated in Step 1 to make the #Problem# more complex. The plan should include several methods from the #Methods List#.
Step 3: Please execute the plan step by step and provide the #Rewritten Problem#. #Rewritten Problem# can only add 10 to 20 words into the "#Problem#".
Step 4: Please carefully review the #Rewritten Problem# and identify any unreasonable parts. Ensure that the #Rewritten Problem# is only a more complex version of the #Problem#. Just provide the #Finally Rewritten Problem# without any explanation and step-by-step reasoning guidance.

Please reply strictly in the following format:
Step 1 #Methods List#:
Step 2 #Plan#:
Step 3 #Rewritten Problem#:
Step 4 #Finally Rewritten Problem#:

#Problem#:  Problem

Figure 11: The prompts used to optimize the difficulty of questions for QPO Training.

---

**Prompts for Problem Solvability Check**

Please act as a professional math teacher.
Your goal is to determine if the given problem is a valuable math problem. You need to consider two aspects:
1. The given problem is a math problem.
2. The given math problem can be solved based on the conditions provided in the problem (You can first try to solve it and then judge its solvability).

Please reason step by step and conclude with either 'Yes' or 'No'.

Given Problem:  Problem

Figure 12: The prompts used to check the solvability of questions.

---

**Prompts for Difficulty Classification**

# Instruction

You first need to identify the given user intent and then label the difficulty level of the user query based on the content of the user query.

## User Query
```
 Input
```

## Output Format
Given the user query, in your output, you first need to identify the user intent and the knowledge needed to solve the task in the user query.
Then, rate the difficulty level of the user query as `very easy`, `easy`, `medium`, `hard`, or `very hard`.

Now, please output the user intent and difficulty level below in a json format by filling in the placeholders in []:
```
{{
"intent": "The user wants to [....]",
"knowledge": "To solve this problem, the models need to know [....]",
"difficulty": "[very easy/easy/medium/hard/very hard]"
}}
```

Figure 13: The prompts used to judge the difficulty level of questions.

---

**Prompts for Topic Classification**

As a mathematics education specialist, please analyze the topics of the provided question and its answer. Specific requirements are as follows:
1. You should identify and categorize the main mathematical topics involved in the problem. If knowledge from non-mathematical fields is used, it is classified into Others - xxx, such as Others - Problem Context.
2. You should put your final answer between <TOPIC> and </TOPIC>.

----

Question: Compute $\cos 330°$.

Answer: We know that $330° = 360° - 30°$.
Since $\cos(360° - \theta) = \cos\theta$ for all angles $\theta$,
we have $\cos 330° = \cos 30°$.
Since $\cos 30° = \frac{\sqrt{3}}{2}$,

we can conclude that $\cos 330° = \boxed{\frac{\sqrt{3}}{2}}$.

Analysis: <TOPIC>Trigonometry - Cosine Function</TOPIC>

----

Question:  Question

Answer:  Answer

Analysis:

Figure 14: The prompts used for topic classification.

**Examples for Solvability Optimization**

**Problems 1 (Before Optimization):**
There are 10 survivors in an emergency room. Each survivor is either a child, a woman, or a man. If there are 4 men and 3 times as many women as men, how many children are there?
**Problems 1 (After Optimization):**
There are 10 survivors in an emergency room. Each survivor is either a child, a woman, or a man. If there are 4 men and an equal number of women as men, how many children are there?

**Problems 2 (Before Optimization):**
How many sides does a polygon have if it is a regular polygon?
**Problems 2 (After Optimization):**
How many sides does a regular polygon have if each interior angle is 120 degrees?

**Problems 3 (Before Optimization):**
Find the sum of the first three terms of this series.
**Problems 3 (After Optimization):**
Calculate the sum of the first three terms of the arithmetic series where the first term is 5 and the common difference is 3.

Figure 15: Three examples for solvability optimization by GPT-4o-mini.

**Examples for Difficulty Optimization**

**Problems 1 (Before Optimization):**
How many 4-digit positive integers are there?
**Problems 1 (After Optimization):**
How many 4-digit positive integers can be formed using non-repeating digits where the sum of these digits must be even, and the integers fall within the range of 1000 to 9999?

**Problems 2 (Before Optimization):**
The average of 15 numbers is 32. An additional number is then added to the list, and the new average of the 16 numbers is 34. What number was added to the list?
**Problems 2 (After Optimization):**
The average of 15 positive integers is 32, but one integer fluctuates to 30 before adding a new number. After adding this new number, the average of the 16 integers becomes 34. Calculate the added number and find the standard deviation of all 16 integers, considering their ascending order.

**Problems 3 (Before Optimization):**
A fair coin is tossed 50 times, what is the probability of getting heads at least 25 times?
**Problems 3 (After Optimization):**
A fair coin is tossed 50 times; what is the probability of obtaining heads at least 25 times, and can you also calculate the expected number, variance, and standard deviation of heads while determining the likelihood that the total number of heads exceeds 30?

Figure 16: Three examples for difficulty optimization by GPT-4o-mini.

**ScaleQuest Example 1**

**QUESTION:**
My orthogonal projection matrix $P$ projects onto the plane with equation

$$x + y + z = 0.$$

Find the matrix $P$.

**SOLUTION:**
To find the orthogonal projection matrix $P$ that projects onto the plane with the equation $x + y + z = 0$, we can follow these steps:
1. **Identify the normal vector to the plane:**

The plane $x + y + z = 0$ has a normal vector $\mathbf{n} = \begin{pmatrix} 1 \\ 1 \\ 1 \end{pmatrix}$.

2. **Express the projection matrix in terms of the normal vector:**
The orthogonal projection matrix $P$ onto the plane can be written as:

$$P = I - \frac{\mathbf{n}\mathbf{n}^T}{\mathbf{n}^T\mathbf{n}}$$

where $I$ is the $3 \times 3$ identity matrix, $\mathbf{n}$ is the normal vector, and $\mathbf{n}^T$ is the transpose of the normal vector.
3. **Calculate $\mathbf{n}^T\mathbf{n}$:**

$$\mathbf{n}^T\mathbf{n} = \begin{pmatrix} 1 & 1 & 1 \end{pmatrix} \begin{pmatrix} 1 \\ 1 \\ 1 \end{pmatrix} = 1^2 + 1^2 + 1^2 = 3$$

4. **Calculate $\mathbf{n}\mathbf{n}^T$:**

$$\mathbf{n}\mathbf{n}^T = \begin{pmatrix} 1 \\ 1 \\ 1 \end{pmatrix} \begin{pmatrix} 1 & 1 & 1 \end{pmatrix} = \begin{pmatrix} 1 & 1 & 1 \\ 1 & 1 & 1 \\ 1 & 1 & 1 \end{pmatrix}$$

5. **Substitute these into the projection matrix formula:**

$$P = I - \frac{1}{3} \begin{pmatrix} 1 & 1 & 1 \\ 1 & 1 & 1 \\ 1 & 1 & 1 \end{pmatrix}$$

where $I$ is the $3 \times 3$ identity matrix:

$$I = \begin{pmatrix} 1 & 0 & 0 \\ 0 & 1 & 0 \\ 0 & 0 & 1 \end{pmatrix}$$

6. **Perform the matrix subtraction:**

$$P = \begin{pmatrix} 1 & 0 & 0 \\ 0 & 1 & 0 \\ 0 & 0 & 1 \end{pmatrix} - \frac{1}{3} \begin{pmatrix} 1 & 1 & 1 \\ 1 & 1 & 1 \\ 1 & 1 & 1 \end{pmatrix}$$

$$P = \begin{pmatrix} \frac{2}{3} & -\frac{1}{3} & -\frac{1}{3} \\ -\frac{1}{3} & \frac{2}{3} & -\frac{1}{3} \\ -\frac{1}{3} & -\frac{1}{3} & \frac{2}{3} \end{pmatrix}$$

Therefore, the orthogonal projection matrix $P$ is:

$$\boxed{\begin{pmatrix} \frac{2}{3} & -\frac{1}{3} & -\frac{1}{3} \\ -\frac{1}{3} & \frac{2}{3} & -\frac{1}{3} \\ -\frac{1}{3} & -\frac{1}{3} & \frac{2}{3} \end{pmatrix}}$$

Figure 17: An example from our constructed dataset.

**ScaleQuest Example 2**

**QUESTION:**
Suppose $a, b$ are positive real numbers such that $a + b = 1$.
(a) minimize $\frac{1}{a} + \frac{1}{b}$;
(b) minimize $a^2 + b^2$.

**SOLUTION:**
To solve these problems, we will use the method of Lagrange multipliers and the AM-HM inequality.
**Part (a):** Minimize $\frac{1}{a} + \frac{1}{b}$ given $a + b = 1$
First, we use the method of Lagrange multipliers. Define the function $f(a, b) = \frac{1}{a} + \frac{1}{b}$ and the constraint $g(a, b) = a + b - 1 = 0$.
The Lagrangian is:
$$\mathcal{L}(a, b, \lambda) = \frac{1}{a} + \frac{1}{b} + \lambda(a + b - 1)$$

Taking the partial derivatives and setting them to zero, we get:
$$\frac{\partial \mathcal{L}}{\partial a} = -\frac{1}{a^2} + \lambda = 0 \implies \lambda = \frac{1}{a^2}$$

$$\frac{\partial \mathcal{L}}{\partial b} = -\frac{1}{b^2} + \lambda = 0 \implies \lambda = \frac{1}{b^2}$$

$$\frac{\partial \mathcal{L}}{\partial \lambda} = a + b - 1 = 0$$

Since $\lambda = \frac{1}{a^2}$ and $\lambda = \frac{1}{b^2}$, we have:
$$\frac{1}{a^2} = \frac{1}{b^2} \implies a^2 = b^2 \implies a = b \quad (\text{since } a, b > 0)$$

Substituting $a = b$ into the constraint $a + b = 1$, we get:
$$a + a = 1 \implies 2a = 1 \implies a = \frac{1}{2} \implies b = \frac{1}{2}$$

Now, we calculate $\frac{1}{a} + \frac{1}{b}$:
$$\frac{1}{a} + \frac{1}{b} = \frac{1}{\frac{1}{2}} + \frac{1}{\frac{1}{2}} = 2 + 2 = 4$$

Thus, the minimum value of $\frac{1}{a} + \frac{1}{b}$ is $\boxed{4}$.
**Part (b):** Minimize $a^2 + b^2$ given $a + b = 1$
We use the identity for the square of a sum:
$$(a + b)^2 = a^2 + b^2 + 2ab$$

Given $a + b = 1$, we have:
$$1^2 = a^2 + b^2 + 2ab \implies 1 = a^2 + b^2 + 2ab \implies a^2 + b^2 = 1 - 2ab$$

To minimize $a^2 + b^2$, we need to maximize $ab$. Using the AM-GM inequality:
$$\frac{a + b}{2} \geq \sqrt{ab} \implies \frac{1}{2} \geq \sqrt{ab} \implies \left(\frac{1}{2}\right)^2 \geq ab \implies \frac{1}{4} \geq ab$$

The maximum value of $ab$ is $\frac{1}{4}$, which occurs when $a = b = \frac{1}{2}$.
Substituting $ab = \frac{1}{4}$ into the expression for $a^2 + b^2$:
$$a^2 + b^2 = 1 - 2ab = 1 - 2\left(\frac{1}{4}\right) = 1 - \frac{1}{2} = \frac{1}{2}$$

Thus, the minimum value of $a^2 + b^2$ is $\boxed{\frac{1}{2}}$.

Figure 18: An example from our constructed dataset.

---

**ScaleQuest Example 3**

**QUESTION:**
Compute

$$\sum_{n=1}^{\infty} \frac{4^n + 5^n + 6^n}{12^n}.$$

**SOLUTION:**
To compute the sum $\sum_{n=1}^{\infty} \frac{4^n+5^n+6^n}{12^n}$, we can split the sum into three separate sums:

$$\sum_{n=1}^{\infty} \frac{4^n + 5^n + 6^n}{12^n} = \sum_{n=1}^{\infty} \frac{4^n}{12^n} + \sum_{n=1}^{\infty} \frac{5^n}{12^n} + \sum_{n=1}^{\infty} \frac{6^n}{12^n}.$$

Each of these sums is a geometric series. Let's evaluate each one separately.
1. For the first sum $\sum_{n=1}^{\infty} \frac{4^n}{12^n}$:

$$\sum_{n=1}^{\infty} \frac{4^n}{12^n} = \sum_{n=1}^{\infty} \left(\frac{4}{12}\right)^n = \sum_{n=1}^{\infty} \left(\frac{1}{3}\right)^n.$$

The sum of an infinite geometric series $\sum_{n=1}^{\infty} ar^{n-1}$ is given by $\frac{a}{1-r}$ for $|r| < 1$. Here, $a = \frac{1}{3}$ and $r = \frac{1}{3}$, so:

$$\sum_{n=1}^{\infty} \left(\frac{1}{3}\right)^n = \frac{\frac{1}{3}}{1 - \frac{1}{3}} = \frac{\frac{1}{3}}{\frac{2}{3}} = \frac{1}{2}.$$

2. For the second sum $\sum_{n=1}^{\infty} \frac{5^n}{12^n}$:

$$\sum_{n=1}^{\infty} \frac{5^n}{12^n} = \sum_{n=1}^{\infty} \left(\frac{5}{12}\right)^n.$$

Here, $a = \frac{5}{12}$ and $r = \frac{5}{12}$, so:

$$\sum_{n=1}^{\infty} \left(\frac{5}{12}\right)^n = \frac{\frac{5}{12}}{1 - \frac{5}{12}} = \frac{\frac{5}{12}}{\frac{7}{12}} = \frac{5}{7}.$$

3. For the third sum $\sum_{n=1}^{\infty} \frac{6^n}{12^n}$:

$$\sum_{n=1}^{\infty} \frac{6^n}{12^n} = \sum_{n=1}^{\infty} \left(\frac{6}{12}\right)^n = \sum_{n=1}^{\infty} \left(\frac{1}{2}\right)^n.$$

Here, $a = \frac{1}{2}$ and $r = \frac{1}{2}$, so:

$$\sum_{n=1}^{\infty} \left(\frac{1}{2}\right)^n = \frac{\frac{1}{2}}{1 - \frac{1}{2}} = \frac{\frac{1}{2}}{\frac{1}{2}} = 1.$$

Adding these three results together, we get:

$$\sum_{n=1}^{\infty} \frac{4^n + 5^n + 6^n}{12^n} = \frac{1}{2} + \frac{5}{7} + 1.$$

To add these fractions, we need a common denominator. The least common multiple of 2, 7, and 1 is 14. So we convert each fraction:

$$\frac{1}{2} = \frac{7}{14}, \quad \frac{5}{7} = \frac{10}{14}, \quad 1 = \frac{14}{14}.$$

Adding these fractions together, we get:

$$\frac{7}{14} + \frac{10}{14} + \frac{14}{14} = \frac{7 + 10 + 14}{14} = \frac{31}{14}.$$

Thus, the sum is:

$$\boxed{\frac{31}{14}}.$$

Figure 19: An example from our constructed dataset.

