# OpenReview forum: "Unleashing Reasoning Capability of LLMs via Scalable Question Synthesis from Scratch"
_ICLR.cc/2025/Conference — Submitted to ICLR 2025_

### Official Review · Reviewer_HR4m · 2024-10-26

**Soundness:** 3
**Presentation:** 3
**Contribution:** 2
**Rating:** 6
**Confidence:** 4

**Summary:**

This paper proposes a synthetic training data generation method for mathematical LLMs. Based on two small models at a 7B scale, the authors achieve state-of-the-art performance than other models trained with the data from larger LMs. The proposed method including question supervised fine-tuning, question preference tuning and reward-score-based selection.

**Strengths:**

As for the method, ScaleQuest generates questions independently from scratch, removing dependency on existing question datasets, which enhances question diversity and supports scalability. Also, the paper integrates comprehensive filtering techniques, including language, solvability, and difficulty sampling, which could be a good reference for future efforts in data filtering.

The presentation is very clear, the workflow of the method is easy to follow. All the details such as prompts are all clearly given. The authors said they will release the data and code, which will be a useful resource to the community.

**Weaknesses:**

The main experiments in Table 1 are somehow not very fair. Some of the baseline methods contain less data than the used dataset in the paper.

In Table 1, it seems that Qwen2-Math-7B-ScaleQuest achieves similar performance with Qwen2-Math-7B-Instruct, I am wondering if their performance is similar on OOD test sets like GSM-hard (https://huggingface.co/datasets/reasoning-machines/gsm-hard) and MathChat (https://github.com/Zhenwen-NLP/MathChat). I would like to see if Qwen2-Math-7B-ScaleQuest is over-fitting on GSM and MATH style questions.

For the efficiency result, it seems that the cost is similar to (even slightly higher) GPT-4o mini if we put that in the table. I am wondering why the authors choose models like Qwen2-Math-7B instead of GPT-4o mini for solvability & difficulty check, etc.

**Questions:**

Typo: Filering -> Filtering in line 215

In Figure 5, it seems that QPO is less effective. Does the author try the combination of QFT and reward filtering only?

I am curious about the effectiveness of Solvability Filtering and Difficulty sampling. For Solvability Filtering, it seems that the final dataset still does not have perfect quality but produces a good performance. So I am curious about the influence of the quality. For difficulty sampling, I am not sure why we need to fit certain difficult distributions.

---

> ### Author Response · Authors · 2024-11-22
> **Response to Reviewer HR4m (Weakness)**
>
> We sincerely appreciate your valuable feedback on our work. Your insights have provided us with an opportunity to refine and improve our paper.
>
> >  To Weakness 1: The main experiments in Table 1 are somehow not very fair.
>
> Exactly, data volume plays a significant role in shaping model performance. Achieving complete fairness in the comparisons, however, is challenging due to certain practical constraints: some datasets, such as WizardMath, MathScale, and KPMath, are closed-source, which limited our access to an equivalent number of training samples.
>
> To ensure fairness, we have taken the following steps:
>
> - For open-source datasets, we plotted the scaling curve in Figure 1 (Right), illustrating the effectiveness of our approach given the same volume of training data.
> - We further provide evaluation results on the same number of training samples (400K) from the public dataset, i.e., MetaMath, MMIQC, DART-Math, and NuminaMath, based on the Qwen2-Math-7B model. More details are shown in Appendix C of our revised version.
>
> |                          | GSM8K    | MATH     | College Math | Olympiad Bench | Avg      |
> | ------------------------ | -------- | -------- | ------------ | -------------- | -------- |
> | Qwen2-Math-7B-MetaMath   | 84.3     | 48.6     | 40.5         | 15.6           | 44.5     |
> | Qwen2-Math-7B-DART-Math  | 88.6     | 58.2     | 45.2         | 22.8           | 52.0     |
> | Qwen2-Math-7B-NuminaMath | 82.0     | 65.8     | 44.9         | 29.2           | 53.7     |
> | Qwen2-Math-7B-ScaleQuest | **90.6** | **71.6** | **50.2**     | **36.2**       | **60.6** |
>
> ---
>
> > To weakness 2: I am wondering if their performance is similar on OOD test sets like GSM-hard, etc
>
> We evaluate results on two other OOD benchmarks, including GSM-hard, MathChat.
>
> | Model                    | GSM-hard | MathChat Follow-up QA      | MathChat Error Correction | Avg  |
> | ------------------------ | -------- | -------------------------- | ------------------------- | ---- |
> | Qwen2-Math-7B-Instruct   | 68.3     | R1: 89.5 R2: 62.4 R3: 53.5 | 89.9                      | 72.7 |
> | Qwen2-Math-7B-ScaleQuest | 66.3     | R1: 89.7 R2: 61.7 R3: 53.5 | 91.1                      | 72.5 |
>
> Our model achieves comparable results to Qwen-Math-7B-Ins, demonstrating its generalization capability.  More details can be seen in Appendix C of our revised version.
>
> ---
>
> > To weakness 3: why the authors choose models like Qwen2-Math-7B, etc
>
> Compared to GPT-4o-mini, Qwen2-Math-7B offers **lower costs** and **more stable and predictable generation times**, which we consider crucial in large-scale data synthesis.
>
> For instance, using GPT-4o-mini to perform solvability checks on 2 million generated questions is estimated to cost around `$600`, and managing such a large volume of API requests introduces additional uncertainty in time consumption. In contrast, using Qwen2.5-Math-7B requires only 110 GPU hours, offering a fixed processing time and significantly reduced costs (~ `$142.7`).

---

> ### Author Response · Authors · 2024-11-22
> **Response to Reviewer HR4m (Question)**
>
> > To Question 1: Typo: Filering -> Filtering in line 215
>
> Thank you for your feedback. We have carefully reviewed and corrected the typo in our manuscript.
>
> ---
>
> > To Question 2: In Figure 5, it seems that QPO is less effective, etc
>
> We would like to correct the misunderstanding that the purpose of QPO is solely to enhance the final SFT model performance. A key objective is actually to improve the efficiency of data generation. As shown in Figure 5, after applying QPO, the solvability of the generated questions increased by 8.2%, a significant improvement, indicating that the sample utilization rate for solvable questions is correspondingly higher.
>
> While the effect may appear minimal due to subsequent solvability filtering, our detailed analysis shows that 28.8% of unsolvable questions were filtered out in the baseline setting, whereas after QPO, only 19.4% were deemed unsolvable. This represents a 9.4% reduction in computational overhead.
>
> ---
>
> > To Question 3: about the effectiveness of Solvability Filtering and Difficulty sampling, etc
>
> We agree that some of the generated questions are not perfect in terms of solvability, which can be attributed to the model’s tendency—driven by hallucination—to attempt solving problems rather than assessing their solvability.
>
> As for difficulty sampling, inspired by DART-Math’s insight that challenging questions can drive more effective learning, we empirically fit specific difficulty distributions by observing the patterns in difficulty distribution.

---

### Official Review · Reviewer_CoxX · 2024-11-02

**Soundness:** 3
**Presentation:** 2
**Contribution:** 2
**Rating:** 6
**Confidence:** 4

**Summary:**

The paper presents ScaleQuest, a scalable and cost-effective data synthesis framework designed to enhance the mathematical problem-solving capabilities of large language models (LLMs). Motivated by the need for high-quality, large-scale data, the authors propose a two-stage synthesis process. Specifically, ScaleQuest employs Question Fine-Tuning (QFT) to activate question-generation (QG) capabilities in small base models and Question Preference Optimization (QPO) to improve question solvability and difficulty. This is followed by filtering for language clarity, difficulty, and solvability, as well as reward-based response selection to ensure high-quality outputs. Experiments demonstrate that models fine-tuned with the ScaleQuest dataset outperform several baselines on benchmarks, achieving substantial improvements in accuracy across in-domain and out-of-domain mathematical reasoning tasks.

**Strengths:**

1. ScaleQuest targets data synthesis for instruction tuning, focusing on affordable low-cost methods. This approach demonstrates significant cost savings (Section 3.4), making large-scale data creation more accessible for open-source communities.

2. The study includes thorough experimentation with multiple baselines, assessing both question and response quality across a total of four mathematical problem-solving benchmarks, thereby increasing the credibility of ScaleQuest.

3. The paper is well-structured and quite easy to follow, with sufficient implementation details to enhance reproducibility.

**Weaknesses:**

1. As claimed by the authors in Lines 17-20 and 76-80, the main contribution of the paper is the scalable synthesis method, ScaleQuest. However, the method heavily depends on domain-specific fine-tuning and specialized models, which raises questions about its generalizability and applicability to domains beyond mathematical reasoning. For instance, the authors use Question Fine-Tuning (QFT) and Question Preference Optimization (QPO) to optimize the question generation process within the target domain of mathematical reasoning. Furthermore, the method involves components like solvability filtering, difficulty sampling, and reward filtering, each relying on different models and a specialized fine-tuned difficulty scorer, which appear tailored to mathematical data construction. This reliance on fine-tuned, domain-specific models, while effective in the tested domain, makes it challenging to adapt ScaleQuest to broader applications, potentially limiting its utility as a general-purpose data synthesis method.

2. Additionally, the paper appears to make some overclaims regarding its scope and efficiency. While the title suggests an enhancement of "reasoning capability," the paper narrowly addresses mathematical reasoning tasks, with little consideration given to other reasoning types, such as causal, commonsense, or logical reasoning. The claim of using “small-size” models (Lines 18-19) is also somewhat misleading. Specifically, the QPO stage (Lines 199-202) requires a larger model, GPT-4o-mini, to achieve better preference-based filtering, suggesting that smaller models alone may not fully support the quality goals of ScaleQuest. The ablation results (Figure 5) further highlight the critical role of QPO, reinforcing the notion that the trade-off between model size and final data quality is not fully acknowledged, which impacts the efficiency claims of the method.

3. Lastly, despite the authors’ assertions that ScaleQuest-generated data significantly enhances performance across various benchmarks, the observed improvements are marginal. For instance, Table 1 shows only a slight average increase from 62.7 to 62.9 when comparing Qwen2-Math-7B-ScaleQuest to its baseline Qwen2-Math-7B-Instruct, even with a decrease in performance on the CollegeMath benchmark. These limited gains suggest that the effectiveness of ScaleQuest’s synthesized data may not justify its complexity. Consequently, these modest gains raise concerns about the practical value and impact of the ScaleQuest approach.

**Questions:**

1. How did the authors select the base difficulty filtering model for fine-tuning (Lines 222-239) and the reward filtering model (Lines 251-252)? Considering that filtering significantly impacts final data quality (Figure 5), further discussion of criteria for model selection, along with any experimental comparisons, would enhance clarity on whether these models represent optimal choices.

2. In Table 1, the term “Synthesis Model” in the header needs clarification. Does it refer to the model used for both question and response generation, or only response generation? This ambiguity is notable, especially as fine-tuned models such as Deepseek-QGen and Qwen2-Math-QGen are absent from the table.

3. The left bar chart in Figure 5 has a confusing y-axis. Does the percentage indicate solvable/non-solvable or easy/difficult ratios? If it reflects these ratios, how does this relate to the five difficulty levels introduced in Lines 377-406? Detailing this connection would make the difficulty and solvability metrics clearer.

4. Lastly, while evaluating synthesized data via difficulty distribution and solvability is helpful, a rigorous human evaluation on a random subset would better demonstrate ScaleQuest’s quality. Including human assessments of clarity, coherence, and real-world relevance could provide a nuanced verification of the synthesized data's effectiveness.

---

> ### Author Response · Authors · 2024-11-22
> **Response to Reviewer CoxX (Weakness)**
>
> > To Weakness 1: This reliance on fine-tuned, domain-specific models, while effective in the tested domain, makes it challenging to adapt ScaleQuest to broader applications, potentially limiting its utility as a general-purpose data synthesis method.
>
> We sincerely apologize for any misunderstandings and would like to clarify that our data generation method in this work is tailored to reasoning tasks, where there is a significant scarcity of high-quality data.
>
> Many general-purpose data generation methods currently fall short in reasoning tasks such as mathematics and code [1, 2]. Reasoning has already become a crucial component of data synthesis [3], with numerous studies dedicated specifically to this area.
>
> Apart from Mathematical Reasoning, we extended our method to the Code Reasoning task as a simple validation, and the results are listed in our response to weakness 2. By keeping the answer generation model and data volume identical, our dataset outperformed the currently popular code dataset, CodeFeedback-Filtered [4]. More details can be seen in Appendix B of the revised version.
>
> [1] https://arxiv.org/abs/2406.08464
>
> [2] https://arxiv.org/abs/2312.15685
>
> [3] https://arxiv.org/abs/2404.07503
>
> [4] https://arxiv.org/abs/2402.14658
>
> ---
>
> >  To Weakness 2: the paper appears to make some overclaims regarding its scope and efficiency, etc
>
> Thank you for your valuable feedback. In response, we included another important task, **code reasoning**, as a simple validation, and the results are as follows:
>
> | Model            |      | HumanEval | MBPP | BigCodeBench | Avg  |
> | ---------------- | ---- | --------- | ---- | ------------ | ---- |
> | CodeFeedback-Raw | 156K | 79.3      | 77.2 | 35.6         | 64.0 |
> | CodeFeedback-Aug | 156k | 84.1      | 84.7 | 39.0         | 69.3 |
> | ScaleQuest-Code  | 156k | 86.6      | 83.1 | 40.0         | 69.9 |
>
> With the same amount of data, our approach outperforms the widely used dataset CodeFeedback-Filtered. Additionally, we augmented the responses for the problems in CodeFeedback-Filtered using the same response generation process as our method, creating a new dataset, CodeFeedback-Aug. The results demonstrate that our approach still achieves superior performance, highlighting the higher quality of the questions generated by our method.
>
> More details are shown in Appendix B in our revised version.
>
> ---
>
> For the QPO stage, smaller models exactly struggle to follow instructions effectively to optimize given questions (as demonstrated in Figure 4, where Qwen2-Math-7B-Ins performs poorly on this task). However, **the data cost for QPO is minimal, requiring only 10K examples**. If human-annotated data were available, it would further enhance the process. GPT-4o-mini was used as a substitute in the absence of available data. Additionally, we experimented with open-sourced Llama3.1-70B-Ins and found that it outperforms GPT-4o-mini in this task.
>
> GPT-4o-mini (solvable ratio): 72.2 -> 83.7
>
> Llama3.1-70B-Ins (solvable ratio): 72.2 -> 86.7
>
> ---
>
> > To Weakness 3: effectiveness of our method, etc
>
> We would like to clarify a misunderstanding: our model is not fine-tuned on Qwen2-Math-7B-Ins but rather on the base model, Qwen2-Math-7B-Base.
>
> Regarding the effectiveness of our method, we would like to emphasize two key points:
>
> - **Qwen2-Math-7B-Ins** demonstrates strong performance on mathematical tasks; however, **none of its training data has been made publicly available**. This means that while the model itself can be used, the underlying data cannot be leveraged for further developments or customized, tailored applications.
> - In contrast, our dataset, generation method, and implementation details are fully open-sourced. While **Qwen2-Math-7B-Ins serves as a teacher model and can be viewed as an "upper bound"**, our work achieves comparable performance and even surpasses it on certain tasks, further demonstrating the effectiveness of our approach.

---

> ### Author Response · Authors · 2024-11-22
> **Response to Reviewer CoxX (Question)**
>
> >  To Question 1: How did the authors select the base difficulty filtering model for fine-tuning (Lines 222-239) and the reward filtering model (Lines 251-252)? etc
>
> - Selection of difficulty filtering model: The selection was inspired by DART-Math, which uses the accuracy of DeepSeek-7B-RL on a given question as a measure of its difficulty. We experimented with different models for training (DSMath-7B-Base and DSMath-7B-RL) and found that the results were similar.
> - Selection of reward filtering model: This choice was primarily guided by the model’s performance on the reasoning subset of the Reward Bench.
>
> Thank you for your suggestion! We have updated the discussion in the revised version.
>
> ---
>
> >  To Question 2: In Table 1, the term "Synthesis Model" in the header needs clarification, etc
>
> Sorry for the confusion. Allow us to clarify:
>
> - DART-Math only includes response generation using DSMath-7B-RL. Other baselines use different synthesis models for both question synthesis and response generation, such as GPT3.5, GPT-4, and GPT-4o.
> - For our approach, DSMath-7B-QGen and Qwen2-Math-7B-QGen are utilized for question synthesis, with Qwen2-Math-7B-Ins used for response generation.
>
> If multiple models are used, only the most recently released one is marked. Additional details regarding the complete synthesis models for these datasets are provided in Figure 6. This clarification has been updated in the caption of our revised version.
>
> ---
>
> > To Question 3: The left bar chart in Figure 5 has a confusing y-axis, etc
>
> Thank you for your suggestion. We have updated the chart with clearer explanations for the solvable ratio and difficulty score. The difficulty score is indeed calculated based on the method described in Lines 377-406, and we have clarified this in the revised version.
>
> ---
>
> > To Question 4: a rigorous human evaluation on a random subset would better demonstrate ScaleQuest’s quality, etc
>
> Thank you for pointing this out. Human evaluation is indeed a more direct approach to demonstrate the quality of ScaleQuest.
>
> We sampled 100 examples each from NuminaMath, and ScaleQuest and evaluated them based on **clarity**, **reasonableness**, and **real-world relevance**, with scores ranging from [1, 5]. (Please understand that due to the complexity of mathematical tasks, we limited the sample size to 40.)
>
> - In terms of clarity and reasonableness, our synthetic data surpasses NuminaMath but still falls short of the high-quality, real-world datasets like the training sets of GSM8K and MATH.
> - Regarding real-world relevance, GSM8K leans toward practical, real-life scenarios, while MATH focuses more on theoretical mathematical derivations. Our generated data can be seen as a balance between the two.
>
> More details have been updated in Appendix C.
>
> |            | clarity | reasonableness | Real-world relevance |
> | ---------- | ------- | -------------- | -------------------- |
> | GSM8K      | 4.4     | 4.5            | 3.9                  |
> | MATH       | 4.1     | 4.3            | 2.4                  |
> | NuminaMath | 3.8     | 4.0            | 2.4                  |
> | ScaleQuest | 3.9     | 4.0            | 2.8                  |

---

> ### Comment · Reviewer_CoxX · 2024-11-24
> **Reviewer’s Comments and Score Update**
>
> The reviewer appreciates the authors’ efforts, particularly the inclusion of additional experiments on code reasoning and human evaluation, which effectively address most of my concerns. Based on these improvements, I have raised my score from 3 to 6 to reflect the enhanced quality of the work. However, I am open to deferring to the consensus of the other reviewers regarding the final decision.

---

### Official Review · Reviewer_KJ61 · 2024-11-03

**Soundness:** 3
**Presentation:** 2
**Contribution:** 2
**Rating:** 5
**Confidence:** 3

**Summary:**

This paper proposes a scalable data synthesis method, ScaleQuest, for math reasoning. The augmented math datasets can enhance the model performance of mainstream open-source models such as Mistral, Llama3, DeepSeekMath, and Qwen2-Math. After finetuning the proposed dataset, the small open-source models can even outperform closed-source models such as GPT-4-Turno and Claude-3.5 Sonnet

**Strengths:**

1. This paper provides a cost-effective data synthesis method for math reasoning problems.
2. The synthetic dataset can boost the performance of multiple open-source models in both in-domain and out-of-domain evaluation.

**Weaknesses:**

1. The main weakness of this paper is, that the proposed data synthesis pipeline is too complex and may be domain-specific. It includes the training in question fine-tuning, question preference optimization, the inference for solvability and difficulty check, reward scoring, etc. Although the API and training cost is not as expensive as GPT-4, this method is more time-consuming and requires extra effort to adapt to other domains.
2. The proposed data synthesis method is only evaluated in the math domain. It is unsure whether this method can be easily adapted to other domains such as code or logical reasoning. Specifically, can the question finetuning and question preference optimization trained on the math domain be directly used for other domains, or the extra finetuning for each domain and each stage is needed?
3. The experimental results are not easy to interpret:
(i) For the baselines with different synthetic datasets, are they finetuned on the same scale of training examples?
(ii) What does the Percentage and Accuracy in Figure 5 mean? Where is the legend of the left plot of Figure 5?
(iii) What does the question quality in Table 2 refer to?
4. There are many components in the data synthesis pipeline, but the impact of each component is not clear. For example, what if removing the question preference optimization and directly using the solvability filtering and difficulty sampling? This is different from the ablation study, which compares the performance w/ and w/o reward filtering while keeping all other components the same.

**Questions:**

There are plenty of LLMs used in the data synthesis pipeline: DeepSeekMath- 7B-RL , Qwen2-Math-7B-Instruct, GPT-4o-mini, GPT-4o, DeepseekMath-7B-Base, InternLM2-7B-Reward. Can you provide a Table for all the settings? Is there any specific reason to select different LLMs for different stages?

---

> ### Author Response · Authors · 2024-11-22
> **Response to Reviewer KJ61 (Weakness 1-3)**
>
> > To Weakness 1: The main weakness of this paper is, that the proposed data synthesis pipeline is too complex and may be domain-specific, etc
>
> Thank you for your feedback. We would like to provide more information about reasoning data synthesis:
>
> - **General-purpose methods struggle with reasoning tasks:** Most general-purpose data generation methods face significant limitations when applied to reasoning tasks, as mentioned in works like Magpie [1].
> - **Reasoning data synthesis requires a more complex process to ensure high-quality data:** Unlike general tasks, reasoning tasks demand higher data quality, which necessitates more sophisticated pipelines like question rephrasing [1], sub-topic extraction[2, 3], verification [1], and quality assessment [3]. Compared to previous works, our approach is more straightforward, only containing QFT, QPO, and filtering process.
> - **Efforts in other reasoning tasks:** We extended our method to the Code Reasoning task as a simple validation, and the results are shown in the figure below. By keeping the answer generation model and data volume identical, our dataset outperformed the currently popular code dataset, CodeFeedback-Filtered [4]. More details can be seen in Appendix B of the revised version.
>
> | Model            |      | HumanEval | MBPP | BigCodeBench | Avg  |
> | ---------------- | ---- | --------- | ---- | ------------ | ---- |
> | CodeFeedback-Raw | 156K | 79.3      | 77.2 | 35.6         | 64.0 |
> | CodeFeedback-Aug | 156k | 84.1      | 84.7 | 39.0         | 69.3 |
> | ScaleQuest-Code  | 156k | 86.6      | 83.1 | 40.0         | 69.9 |
>
> [1] https://arxiv.org/abs/2309.12284
>
> [2] https://arxiv.org/abs/2403.02333
>
> [3] https://arxiv.org/abs/2403.02884
>
> [4] https://arxiv.org/abs/2402.14658
>
> ---
>
> > To Weakness 2: The proposed data synthesis method is only evaluated in the math domain.
>
> Thank you for your valuable feedback. To address this, we made minor adjustments to the ScaleQuest method to adapt to code reasoning tasks (details in Appendix B). Our experiments demonstrate that the resulting dataset achieves higher quality compared to the open-source CodeFeedback dataset. The results are provided above (see response in Weakness 1), with additional details included in Appendix B.
>
> ---
>
> >  To Weakness 3.1: For the baselines with different synthetic datasets, are they finetuned on the same scale of training examples?
>
> We apologize for the confusion. The results in Table 1 do not strictly control for identical training data volumes due to practical constraints (e.g., some datasets are not publicly available).
>
> To ensure a fair comparison, we made the following efforts:
>
> - For open-source datasets, we plotted the scaling curve in Figure 1 (Right), which demonstrates the effectiveness of our approach with the same volume of training data.
> - Additionally, we provide evaluation results using Qwen2-Math-7B fine-tuned on 400K training samples drawn from public datasets (MetaMath, DART-Math, and NuminaMath). The new results are updated in Appendix C.
>
> |                          | GSM8K | MATH | College Math | Olympiad Bench | Avg  |
> | ------------------------ | ----- | ---- | ------------ | -------------- | ---- |
> | Qwen2-Math-7B-MetaMath   | 84.3  | 48.6 | 40.5         | 15.6           | 44.5 |
> | Qwen2-Math-7B-DART-Math  | 88.6  | 58.2 | 45.2         | 22.8           | 52.0 |
> | Qwen2-Math-7B-NuminaMath | 82.0  | 65.8 | 44.9         | 29.2           | 53.7 |
> | Qwen2-Math-7B-ScaleQuest | 90.6  | 71.6 | 50.2         | 36.2           | 60.6 |
>
> ---
>
> >  To Weakness 3.2: What does the Percentage and Accuracy in Figure 5 mean? Where is the legend of the left plot of Figure 5?
>
> We are sorry for the confusion,
>
> - **Percentage:** In the solvability subplot, it indicates the proportion of generated questions judged as solvable. In the difficulty subplot, it represents the average difficulty score of the generated questions.
> - **Accuracy:** This metric evaluates the impact of the synthesized dataset on model performance, specifically the fine-tuned model’s accuracy on the test dataset. Detailed explanations can be found in Section 3.1 under *Evaluation and Metrics.*
>
> We have clarified these definitions in the figure caption of our revised version.
>
> ---
>
> > To Weakness 3.3: What does the question quality in Table 2 refer to?
>
> We are sorry for the confusion. "question quality" in Table 2 refers to instruction tuning effectiveness.
>
> The purpose of Table 2 is to compare the instruction tuning effectiveness of questions from different open-source datasets. To ensure a fair comparison, we kept the answer generation process entirely identical.

---

> ### Author Response · Authors · 2024-11-22
> **Reponse to Reviewer KJ61 (Weakness 4 & Question)**
>
> > To Weakness 4: There are many components in the data synthesis pipeline, but the impact of each component is not clear, etc
>
> We would like to clarify the purpose of our ablation study, which is to demonstrate the effectiveness of each submethod in the pipeline. Our ablation results, as shown in Figure 5, include evaluations with **solvability** and **difficulty filtering**, highlighting their contributions.
>
> More analysis has been updated in Appendix C.
>
> | +QFT | +QPO | +RF  | Avg of 4 Benchmark |
> | ---- | ---- | ---- | ------------------ |
> | No   | No   | No   | 42.2               |
> | No   | No   | Yes  | 44.1               |
> | Yes  | No   | No   | 52.9               |
> | Yes  | Yes  | No   | 53.3               |
> | Yes  | No   | Yes  | 54.7               |
> | Yes  | Yes  | Yes  | 55.1               |
>
> ---
>
> > To Question: There are plenty of LLMs used in the data synthesis pipeline: DeepSeekMath-7B-RL , Qwen2-Math-7B-Instruct, GPT-4o-mini, GPT-4o, DeepseekMath-7B-Base, InternLM2-7B-Reward. Can you provide a Table for all the settings? Is there any specific reason to select different LLMs for different stages?
>
> | Stage                         | Models                          | purpose                                                      | Why to choose                                                |
> | ----------------------------- | ------------------------------- | ------------------------------------------------------------ | ------------------------------------------------------------ |
> | Train Question Generator      | DSMath-7B-RL; Qwen2-Math-7B-Ins | Activate the question generation capability of problem-solving models. | Two recent problem-solving models; Multiple generators contribute to greater data diversity. |
> | Query Preference Optimization | GPT-4o-mini; GPT-4o             | Construct preference data                                    | GPT-4o-mini is better suited for following instructions to optimize questions (as shown in Figure 4); GPT-4o is used to judge whether generated questions are solvable. |
> | Solvability filtering         | Qwen2-Math-7B-Ins               | Check the solvability of each generated question.            | Large-scale generated data should be processed, so the 7B-scale model should be a cheaper choice. |
> | Difficulty filtering          | DSMath-7B-base                  | Generate score for each question                             | Inspired by DART-Math, we experimented with DSMath-7B-Base and DSMath-7B-RL and found only a minimal difference between them. |
> | Response Generation           | Qwen2-Math-7B-Ins               | Generate response                                            | Recent Math Problem-Solving Model.                           |
> | Reward Filtering              | InternLM-7B-Reward              | Assign a reward score for each response and keep the one with the highest score. | This choice is primarily based on the model’s performance on the reasoning subset of the Reward Bench. |

---

> > ### Comment · Reviewer_KJ61 · 2024-11-23
> >
> > Thanks for the authors' efforts in rebuttal. The revised manuscript addresses my concerns about the unclear descriptions of experimental results and ablation studies.
> >
> > However, after reading the response, I would like to maintain my score for the following reasons:
> >
> > 1. *The adaptation capability of the proposed method to a new domain is cost-consuming and ineffective.*  The new results on code generation do not show a significant improvement over the existing datasets. Moreover, the adaptation requires retraining two question generation models, which is costly. The question preference model is reused for code, but their effectiveness is questionable due to the large difference between the math and the code questions.
> > 2. *The proposed pipeline is too complex with unnecessary complexity in the choice of LLMs.*  From the table of LLMs' settings, I do not think it is necessary to use so many different kinds of LLMs. For example, why not use the same Qwen2-Math-7B-Ins for Train Question Generator, Solvability filtering, Difficulty filtering, and Response Generation? Why not use GPT-4o-mini for both generation and evaluation if GPT-4o is too expensive? This makes it unclear whether the improvement comes from the pipeline design or the specific choice of LLMs. It also makes it more difficult to apply the proposed pipeline to a new domain effectively. For example, how to choose the correct LLM for each stage when transferring to the code domain? Is the current limited improvement mainly because the selection of LLMs is not optimal?

---

> ### Author Response · Authors · 2024-11-25
> **Further Clarifications and Experiments**
>
> Thanks for the quick feedback. We would like to clarify some unclear points in our first round of the response. We hope the newly attached response and experiments mitigate some of your concerns.
>
> > To Concern 1: The adaptation capability of the proposed method to a new domain is cost-consuming and ineffective.
>
> We apologize for any confusion regarding the settings. To address the concern about effectiveness in code reasoning task, we would like to provide further clarification based on the experimental results for code reasoning:
>
> - **CodeFeedback-Raw** refers to the public version, which consists of a condensed collection of problems from various open-source datasets. It is one of the most frequently used datasets for code reasoning. Our method demonstrates significant improvements over this dataset (5.3% in average accuracy).
> - **CodeFeedback-Aug**: To ensure a fair comparison, we applied the same reward filtering method to enhance the responses of existing problems, resulting in CodeFeedback-Aug. The quality of the problems synthesized by our approach is comparable to, or even surpasses, that of CodeFeedback itself, which includes both **real-world questions and high-quality synthetic questions**. This result further highlights the high quality of the questions generated by our method.
>
> ---
>
> Regarding the concern about cost-effectiveness, we would like to provide the following clarifications:
>
> - The training cost for the question generator is minimal, requiring only around 20 GPU hours (approximately 2.5 hours on an 8 A100-40G-PCIe server), which constitutes just 3% of the total data synthesis cost, as shown in Table 4. While using a single question generator is entirely feasible, employing multiple generators yields better results, as discussed in *Multiple question generators enhance data diversity* in Section 3.3. We also simplify our approach and the results can be seen in our following response to concern 2.

---

> > ### Author Response · Authors · 2024-11-25
> > **Continued (Response to Concern 2)**
> >
> > > To concern 2: The proposed pipeline is too complex with unnecessary complexity in the choice of LLMs, etc
> >
> > Thank you for your feedback. We acknowledge and apologize for the confusion caused by directly presenting a complex setup. The approach we initially presented represents a culmination of all our insights, which were systematically validated through the subsequent ablation studies.
> >
> > To address this, we have supplemented our revised version with a simpler setup to eliminate potential misunderstandings for readers. We used Qwen2-Math-7B-Ins for training question generators, constructing optimization data for QPO, performing solvability & difficulty filtering, as well as for response generation. For reward filtering, InternLM-7B-Reward remained unchanged. The results, as shown below (ScaleQuest-Simple), indicate that our approach continues to demonstrate superior performance compared to existing datasets. The corresponding results and analysis have been included in Appendix C of the revised version.
> >
> > |                                  | Samples | GSM8K    | MATH     | College Math | Olympiad Bench | Average  |
> > | -------------------------------- | ------- | -------- | -------- | ------------ | -------------- | -------- |
> > | Qwen2-Math-7B-MetaMath           | 395K    | 84.3     | 48.6     | 40.5         | 15.6           | 47.3     |
> > | Qwen2-Math-7B-DART-Math          | 400K    | 88.6     | 58.2     | 45.2         | 22.8           | 53.7     |
> > | Qwen2-Math-NuminaMath            | 400K    | 82.0     | 65.8     | 44.9         | 29.2           | 55.5     |
> > | Qwen2-Math-ScaleQuest            | 400K    | **90.6** | **71.6** | **50.2**     | **36.2**       | **62.1** |
> > | **Qwen2-Math-ScaleQuest-simple** | 400K    | 89.4     | 69.9     | 48.8         | 33.6           | 60.4     |
> >
> > ---
> >
> > Regarding GPT-4o, it is solely used as a manual substitute to evaluate solvability and difficulty for demonstrating the effectiveness of question preference optimization, and it is not involved in the overall ScaleQuest process. Additionally, we have included human evaluations, detailed in Appendix C, to further validate the effectiveness of our approach.
> >
> > ---
> >
> > Additionally, we believe it is worthwhile to summarize these insights on model selection for domain adaptation in the revised version:
> >
> > - **Selection of base model for training question generator:** The self-synthesis generation paradigm heavily relies on the inherent knowledge of the problem-solving model itself [1]. Therefore, a domain-specific model is essential. For example, Qwen2-Math-Ins is suitable for mathematical reasoning, while Qwen2.5-Coder-Ins fits well for code reasoning. Furthermore, using multiple question generators often leads to more diverse and higher-quality questions (as discussed in section 3.3).
> > - **Selection of model for constructing optimization data:** Well-aligned, general-purpose models, such as Llama3.1-70B and GPT-4o-mini, tend to perform better than domain-specific models, as illustrated in Figure 4.
> > - **Selection of Response Generation Model & Reward Model:** These can be selected based on their performance on the corresponding benchmarks, like MATH for mathematical reasoning, BigCodeBench for code reasoning, and Reward Bench for the reward model. We will also discuss the impact of different reward models, and the final results will be updated before the end of the author response period.
> >
> > In fact, we did not fully utilize the most advanced and optimal models available. For instance, Qwen2.5-Math-72B-Ins would be a better choice for training the question generator and serving as a response generator, while Qwen2.5-Math-72B-Reward is undoubtedly a superior option for reward filtering.
> >
> > Moreover, math and code are two representative and non-trivial areas of focus within reasoning tasks [2]. To adapt to other domains, modifications should involve: (1) domain-specific problem-solving models for QFT and response generation, and (2) an optimization prompt tailored to the domain. More fine-grained optimization designs in QPO and the use of domain-specialized reward models (like Qwen2.5-Math-72B-Reward for Math Domain) could further improve performance. **Overall, We believe that leveraging the key insights outlined above can facilitate a straightforward adaptation to other reasoning domains.**
> >
> > We also believe that **the methodology and the experience in selecting models are always more critical than the chosen models themselves**. With the continuous advancements in the open-source community, we are confident that stronger models will undoubtedly produce even better datasets when applying our approach.
> >
> >
> >
> > [1] https://arxiv.org/abs/2406.08464
> >
> > [2] https://arxiv.org/abs/2404.07503

---

### Official Review · Reviewer_FQiU · 2024-11-04

**Soundness:** 3
**Presentation:** 2
**Contribution:** 2
**Rating:** 5
**Confidence:** 5

**Summary:**

The paper introduces a novel framework for generating high-quality reasoning datasets using smaller open-source models. The primary focus is on addressing the challenges of synthesizing high-quality data at scale with affordable costs.

Key contributions of the paper include:

- The authors present a scalable data synthesis method that enables the generation of 1 million question-answer pairs without relying on extensive seed data or complex augmentation techniques.

- The framework incorporates a two-stage process consisting of Question Fine-Tuning (QFT) and Question Preference Optimization (QPO), which enhances the question generation capabilities of the models.

- The paper demonstrates that models fine-tuned with the ScaleQuest dataset achieve significant performance gains compared to baseline models.

**Strengths:**

- This article focuses on synthesizing mathematical problems using open-source large language models, which is an important topic. The fine-tuning and filtering techniques proposed by the authors demonstrate some effectiveness.
- The article presents a thorough and detailed set of experiments.

**Weaknesses:**

- The proposed Question Preference Optimization (QPO) appears to be less effective; as shown in Figure 5, the difference between QPO and QFT is minimal, raising questions about the validity of QPO.

- This paper attempts to extract training data from models, similar to the approach of MAGPIE. Therefore, the authors should conduct a more fair and detailed comparison between Question Fine-Tuning (QFT) and direct prompting methods. In Figure 5, the authors generated 1 million question-response pairs using MAGPIE with Qwen2-Math-7B-Instruct as the "raw data" setting. However, the other settings filtered 2 million( from DeepSeekMath-QGen and Qwen2-Math-QGen) questions down to 1 million and applied a reward model to filter the responses. Consequently, it is difficult to determine whether QFT is more effective than the MAGPIE method or if the filtration of questions and responses is more effective.

- The ablation experiments are insufficient. The authors conducted experiments only on Llama3-8B, rather than comparing all four base models as presented in the main table.

- The authors suggest that the data generation method proposed in this paper can produce diverse and high-quality questions at a lower cost. However, with advancements in open-source models, previous sample-driven and knowledge-driven question synthesis models can also be replaced with open-source models. Moreover, Qwen2-Math, as a response synthesizer, demonstrates superior mathematical capabilities compared to earlier versions of GPT-4. Therefore, it is difficult to assert that the data synthesis approach presented in this paper is superior to other methods in cost.

**Questions:**

- The authors should compare different base models in Figure 5 and Table 2.

- The experimental setup in the experimental module should be clearly presented; for instance, in Table 2, did the responses corresponding to questions from other datasets involve generating five responses and filtering down to one based on the reward model, or was only one response generated?

- The authors might discuss the effects of optimizing different question data volumes during QPO. Additionally, since the authors note that optimizing for both solvability and difficulty simultaneously in QPO is challenging, are there corresponding experimental results to support this?

- The author should probably compare the generated questions with the questions in the test set (n-grams or other methods) to prevent potential data leakage.

---

> ### Author Response · Authors · 2024-11-22
> **Response to Reviewer FQiU (Weakness)**
>
> > To Weakness 1: The proposed Question Preference Optimization (QPO) appears to be less effective; as shown in Figure 5, the difference between QPO and QFT is minimal, raising questions about the validity of QPO.
>
> We agree that the impact of QPO on the final SFT model’s performance is limited. Allow us to provide further clarification:
>
> - **Regarding the limited SFT performance:** QPO specifically targets question generation and does not directly affect the response generation process, which explains the relatively modest performance gains. However, our results demonstrate that QPO is effective, achieving consistent improvements in the solvability and difficulty of generated questions, as well as in the overall SFT effectiveness.
> - **Beyond SFT performance:** Though the impact of QPO on SFT may be minimal, it significantly enhances the **data generation efficiency.** Specifically, QPO improves the solvability of generated questions from 75.4% to 83.6%, a meaningful enhancement that boosts the efficiency of data utilization. While the effect may appear minimal due to subsequent solvability filtering, our detailed analysis shows that 28.8% of unsolvable questions were filtered out in the baseline setting, whereas after QPO, only 19.4% were deemed unsolvable. This represents a 9.4% reduction in computational overhead.
>
> ---
>
> > To Weakness 2: the authors should conduct a more fair and detailed comparison, etc
>
> Thank you for pointing out this unfair setting. To ensure a fair comparison and demonstrate the effectiveness of QFT, we controlled the question generator to be consistent (using DSMath-7B-RL and Qwen2-Math-7B-Ins, each generating 1M data points, i.e., **2M in total for Magpie**). We then applied language, solvability, and difficulty filtering, followed by response generation and reward filtering. This process resulted in approximately 1.2M data points for SFT. The results (based on Qwen2-Math-7B) are as follows:
>
> | Method | GSM8K | MATH | College Math | Olympiad Bench | Avg  |
> | ------ | ----- | ---- | ------------ | -------------- | ---- |
> | Magpie | 75.9  | 47.7 | 38.2         | 14.6           | 44.1 |
> | Ours   | 89.7  | 73.4 | 50.0         | 38.5           | 62.9 |
>
> The corresponding results have been updated in the revised version (Figure 1 and Figure 5).
>
> ---
>
> > To Weakness 3: The ablation experiments are insufficient, etc
>
> Thank you for your feedback, and we sincerely apologize for any shortcomings in our ablation experiments. We have added Figure 5 to include results using Qwen2-Math-7B as a base model, providing a broader comparison.
>
> We hope you understand that the SFT process is quite time-consuming for us. We will try to complete the experiments for the remaining two base models and include the results before the end of rebuttal period.
>
> ---
>
> > To Weakness 4: with advancements in open-source models, previous sample-driven and knowledge-driven question synthesis models can also be replaced with open-source models, etc
>
> Exactly, more advanced models tend to generate better questions and answers. Here are our key observations:
>
> **Limited effectiveness of open-source models in question sample/knowledge-driven approaches:** Some math-specialized models, such as Qwen2-Math-7B, are not well-suited for sample-driven or knowledge-driven approaches. These methods demand the model to generate valuable questions under complex constraints (e.g., specific topics or knowledge points), a task that these problem-solving models often struggle with. For example, when we used Qwen2-Math-7B-Ins to optimize given questions, the quality of the generated questions even declined (as shown in the top plot of Figure 4). Similarly, general-purpose models like Llama3-8B also faced challenges in producing high-quality questions due to their lack of mathematical specialization.
>
> This may be why many sample-driven and knowledge-driven approaches rely on highly aligned closed-source models like GPT-4. This also highlights the unique value of ScaleQuest in enabling open-source models to overcome these limitations.
>
> **Efficiency advantages of our approach:** Our question generation process requires only a minimal number of tokens (e.g., a BOS token), and the generated content directly serves as the final question without any redundant steps. In contrast, sample-driven or knowledge-driven methods often rely on heavily constrained prompts and multiple rounds of verification. **This will result in significantly higher consumption of input and output tokens, leading to greater computational overhead, regardless of the model used**.

---

> > ### Comment · Reviewer_FQiU · 2024-11-27
> > **Official Comment by Reviewer FQiU**
> >
> > Thanks to the author for his detailed reply to my concern.
> >
> > - Judging from the additional experiments, the improvement brought by the QPO design is too small. Maybe it is even better to generate more questions in the question generation stage?
> >
> > - As can be seen from Figure 5, MAGPIE has also improved to a certain extent after passing Question and Response Filter, but it is still lower than the author's method. If the generation conditions (sampling coefficients) are consistent, the method proposed in this article can indeed be better than MAGPIE. Efficiently extract training data from the model.
> >
> > Overall, the QFT proposed by the author has certain effectiveness on synthetic data (QPO effects are too small, and FIlter technology is existing).
> >
> > I will adjust the soundness from 2 to 3, but I still think that the technical contribution of this article cannot meet the acceptance threshold of this conference.

---

> > > ### Author Response · Authors · 2024-11-28
> > > **Further Clarification Regarding Your Concern**
> > >
> > > Thank you for your valuable feedback.
> > >
> > > Increasing the amount of data can indeed lead to some improvement, as we discuss in Appendix C of our revised version. However, the extent of this improvement is limited. This experience also aligns with the that of DPO when applied to response optimization. We believe that 10k–15k training samples are a reasonable and balanced choice.
> > >
> > > ---
> > >
> > > The main concern lies in the effectiveness of QPO.
> > >
> > > We would like to clarify that QPO is not the core technical contribution of our work; rather, it constitutes only a small part of our overall method, and we have not overclaimed QPO as a core contribution. ScaleQuest is a comprehensive framework comprising multiple components, and we are pleased to see that the effectiveness of other sub-methods, such as QFT, solvability analysis, and reward filtering, has been recognized.
> > >
> > > In addition to offering technical contributions, we believe that proposing promising directions to address significant problems and tasks is equally important for the acceptance of this conference.
> > >
> > > We would like to clarify the contributions of our work as follows:
> > >
> > > - **Overall Contribution**: To the best of our knowledge, we are the first to propose the concept of "**Training a Question Generator**" for reasoning tasks, which we consider a promising direction for future research. We introduced QFT (Question Fine-Tuning) and QPO (Question Preference Optimization), which correspond to traditional instruction and preference tuning. While the effectiveness of QPO is currently limited, we demonstrate that questions themselves can be optimized to improve solvability, difficulty, and instruction tuning effectiveness. This is a promising direction worth further exploration, with significant potential for improvement. We believe that a more refined design for QPO could lead to greater enhancements. However, developing such sophisticated algorithms is beyond the primary focus of this paper, and we leave the concentrated study of question preference tuning for future work.
> > > - **Effectiveness**: Our approach effectively tackles the challenge of data generation for reasoning tasks, demonstrating significant improvements over existing math-specialized models, as illustrated in Figure 1. Specifically, our data delivers a 46.4% accuracy improvement for Mistral-7B, 43.2% for Llama3-8B, 31.1% for DSMath-7B, and 29.2% for Qwen2-Math. Overall, our framework is both cost-efficient and our data ranks among the highest-quality open-source datasets for instruction tuning.
> > > - **Contribution to open-source community**: Previous works heavily rely on the strong instruction-following capabilities of closed-source models like GPT-4, with some failing to publicly release their datasets and code. In contrast, our framework is fully open-source, allowing for the generation of data with both high quality and diversity. Furthermore, we have adapted our method to code reasoning tasks, with promising results detailed in Appendix B.
> > >
> > > We hope the above clarifications will help you reassess our contribution with a fresh perspective.

---

> > > > ### Comment · Reviewer_FQiU · 2024-11-29
> > > > **Official comment**
> > > >
> > > > Thanks for your Clarification
> > > >
> > > >
> > > > "Training a Question Generator" is not the firstly proposed by this paper. For example, [1] synthesizes 6 million math problems from their trained question generator.  Also, I think training a question generator is not a very novel idea and only requires simple SFT techniques.
> > > >
> > > >
> > > > [1] JiuZhang3.0: Efficiently Improving Mathematical Reasoning by Training Small Data Synthesis Models

---

> > > > > ### Author Response · Authors · 2024-11-29
> > > > > **Author Responses**
> > > > >
> > > > > Thanks for your feedback.
> > > > >
> > > > > Regarding your comment on "training a question generator", we would like to provide the following clarifications:
> > > > >
> > > > > - In Jiuzhang 3.0, the "question generator" can be seen as a distillation version of GPT-4 (as mentioned in their abstract: "*we create a dataset using GPT-4 to distill its data synthesis capability into the small LLM*"). The primary focus is on "distillation", whereas our concept of "training" goes beyond the limitations of the teacher model, offering the potential for a higher performance ceiling.
> > > > > - More importantly, we believe it is more accurate to refer to such models as "**question extractors**" rather than "question generators", as it extracts potential valuable questions from large-scale pretraining data. This extractive approach has already been explored in previous works [1,2,3]. The extracted data often (1) heavily depends on pretraining corpora and (2) lacks high quality, which limits its applicability for fine-grained instruction fine-tuning (IFT) in recent works.
> > > > >
> > > > > Regarding the concept of "training a question generator" proposed in our paper, we would like to explain as follows:
> > > > >
> > > > > - The "question generator" for IFT should be capable of generating questions from scratch [4] or based on specific topics or knowledge [5, 6], which has been demonstrated to produce higher-quality and more diverse data for Instruction Fine-Tuning.
> > > > > - Similar to Jiuzhang 3.0, GPT-4 may still be a better choice if cost and closed-source limitations are ignored. However, this highlights the significance of our work: we propose a fully open-source solution, which we see as a valuable contribution to the open-source community.
> > > > > - While, as you mentioned, our method might not excel in finer design aspects like QPO, we believe its potential for further improvements has been acknowledged from your questions. From this perspective, our work can serve as a strong baseline for future research. We also believe that the eventual success of alternative methods or models will build upon this recognized "potential".
> > > > >
> > > > > Thank you again for your feedback. We hope the above clarifications address your concerns, and look forward to further discussions.
> > > > >
> > > > > [1] Instruction Pre-Training:Language Models are Supervised Multitask Learners
> > > > >
> > > > > [2] Augmenting Math Word Problems via Iterative Question Composing
> > > > >
> > > > > [3] JiuZhang3.0: Efficiently Improving Mathematical Reasoning by Training Small Data Synthesis Models
> > > > >
> > > > > [4] Alignment Data Synthesis from Scratch by Prompting Aligned LLMs with Nothing
> > > > >
> > > > > [5] MathScale: Scaling Instruction Tuning for Mathematical Reasoning
> > > > >
> > > > > [6] Key-Point-Driven Data Synthesis with its Enhancement on Mathematical Reasoning

---

> ### Author Response · Authors · 2024-11-22
> **Response to Reviewer FQiU (Question)**
>
> > To Question 1: The authors should compare different base models in Figure 5 and Table 2.
>
> Please see our response to weakness 3.
>
> ---
>
> > To Question 2: The experimental setup in the experimental module should be clearly presented, etc
>
> We apologize for the confusion caused by our unclear descriptions. We have carefully reviewed and clarified the setup for each experiment in our revised version.
>
> - For Table 2, we ensured that the response generation process was consistent.
> - For Figure 5, we added the explanation of the solvable ratio and difficulty score.
>
> ---
>
> > To Question 3: The authors might discuss the effects of optimizing different question data volumes, etc
>
> We have further explored the impact of varying training data volumes on QPO. Using Qwen2-Math-7B-Ins as an example, we conducted experiments with 5K, 10K, 15K, 20K, and 40K samples. The results are presented below, and we discuss them in detail in Appendix C of our revised version.
>
> | Train Data | Solvable Ratio | Difficulty Score |
> | ---------- | -------------- | ---------------- |
> | 0K         | 75.8           | 49.6             |
> | 5K         | 81.5           | 50.8             |
> | 10K        | 83.8           | 50.9             |
> | 15K        | 84.5           | 50.7             |
> | 20K        | 84.9           | 50.9             |
> | 40K        | 85.2           | 51.0             |
>
> ---
>
> > To Question 4: The author should probably compare the generated questions with the questions in the test set (n-grams or other methods) to prevent potential data leakage.
>
> We appreciate the reviewer's concern about potential data leakage. To address this, we have conducted an n-gram similarity analysis between the generated questions and all test sets from both our dataset and other baseline datasets. Based on prior empirical analysis [1, 2], we set n=13 to prevent spurious collisions and calculated how much the test sets overlap with training data to assess data contamination. The table below illustrates the clean ratio across our dataset and baseline datasets, defined as the percentage of test samples containing no n-gram matches with the training set. The experiments and analysis have been updated in Appendix A of our revised version.
>
> | Train Data | GSM8K  | MATH   | College Math | Olympiad Bench | Average |
> | ---------- | ------ | ------ | ------- | -------- | ------- |
> | MetaMath   | 99.77% | 92.20% | 100.00% | 99.70%   | 97.92%  |
> | NuminaMath | 99.77% | 89.76% | 99.86%  | 86.81%   | 94.05%  |
> | DartMath   | 99.77% | 91.46% | 100.00% | 99.56%   | 97.70%  |
> | MMIQC      | 99.77% | 88.04% | 98.90%  | 97.93%   | 96.16%  |
> | SacleQuest | 99.85% | 92.82% | 99.75%  | 97.19%   | 97.40%  |
>
> The results demonstrate that our dataset achieves a relatively high level of data cleanliness compared to other datasets, suggesting that our method generates novel questions instead of memorizing existing ones.
>
> [1] https://arxiv.org/abs/2005.14165
>
> [2] https://arxiv.org/abs/2109.01652

---

> > ### Author Response · Authors · 2024-11-25
> > **Additional Experiments**
> >
> > We have completed the inclusion of all four base models in Table 2, as referenced in Weakness 3 and Question 1. The results have been updated in Appendix C of our revised version.
> >
> > Using Mistral-7B as the base model:
> >
> > | Method                | GSM8K    | MATH     | College Math | Olympiad Bench | Average  |
> > | --------------------- | -------- | -------- | ------------ | -------------- | -------- |
> > | Mistral-7B-MetaMath   | 77.0     | 34.1     | 18.6         | 8.6            | 34.6     |
> > | Mistral-7B-OrcaMath   | 84.4     | 31.6     | 20.9         | 8.2            | 36.3     |
> > | Mistral-7B-NumiMath   | 79.5     | 62.8     | 40.4         | **30.4**       | 53.3     |
> > | Mistral-7B-ScaleQuest | **88.5** | **62.9** | **43.5**     | 28.8           | **55.9** |
> >
> > Using Llama3-8B as the base model:
> >
> > |                      | GSM8K    | MATH     | College Math | Olympiad Bench | Average  |
> > | -------------------- | -------- | -------- | ------------ | -------------- | -------- |
> > | Llama3-8B-MetaMath   | 77.6     | 33.1     | 20.6         | 9.2            | 35.1     |
> > | Llama3-8B-OrcaMath   | 83.2     | 32.6     | 19.4         | 8.6            | 36.0     |
> > | Llama3-8B-NumiMath   | 79.1     | 62.9     | 39.3         | **25.4**       | 51.7     |
> > | Llama3-8B-ScaleQuest | **87.9** | **64.4** | **42.8**     | 25.3           | **55.1** |
> >
> > Using Qwen2-Math-7B as the base model:
> >
> > | Method                   | GSM8K    | MATH     | College Math | Olympiad Bench | Average  |
> > | ------------------------ | -------- | -------- | ------------ | -------------- | -------- |
> > | Qwen2-Math-7B-MetaMath   | 88.5     | 68.5     | 47.1         | 33.0           | 59.3     |
> > | Qwen2-Math-7B-OrcaMath   | 89.3     | 68.3     | 46.6         | 31.9           | 59.0     |
> > | Qwen2-Math-7B-NumiMath   | 89.5     | 72.6     | 49.5         | 36.3           | 62.0     |
> > | Qwen2-Math-7B-ScaleQuest | **89.7** | **73.4** | **50.0**     | **38.5**       | **62.9** |

---

### Author Response · Authors · 2024-11-22
**General Response**

We sincerely thank all reviewers for the thoughtful and constructive feedback, as well as the time and effort in reviewing our work. The insights have been invaluable in helping us refine and improve our work.

In response to the feedback, we have submitted a revised version of the manuscript with the following major updates:

- **Extension to another reasoning task:** We validated our approach on an additional reasoning task, *code reasoning*, as suggested by Reviewers KJ61 and CoxX. Our dataset outperformed the popular open-source dataset CodeFeedback, further demonstrating the effectiveness of our method. Details can be found in Appendix B.
- **Enhanced evaluations and analysis to validate effectiveness:** We conducted additional ablation studies and analyses, including **fair comparison experiments (same volume of training data)**, results on more base models, the impact of different training data volumes on QPO, additional results on OOD benchmarks, and human evaluations of the generated dataset. Further details are provided in Appendix C.
- **Corrections and clarifications:** We addressed typos and provided clearer explanations of the experimental setup to enhance understanding and reproducibility.

We are grateful for these insights, which have significantly contributed to improving the quality of our work.

---

### Meta-Review · Area_Chair_iwSp · 2024-12-19

**Metareview:**

This paper presents ScaleQuest, a framework for generating high-quality mathematical reasoning datasets using smaller open-source models. The authors propose a two-stage process combining Question Fine-Tuning (QFT) and Question Preference Optimization (QPO) to enable generation of diverse math problems without extensive seed data. Using this approach, they generated 1 million problem-solution pairs and demonstrated that models fine-tuned on this dataset achieved substantial improvements of 29.2% to 46.4% on the MATH benchmark, with competitive performance against larger proprietary models.

While the paper addresses an important problem of making training data generation more accessible and cost-effective, there are several critical limitations that warrant rejection. First, the proposed pipeline is unnecessarily complex, involving multiple stages and different models without clear justification for these design choices. This complexity not only makes the method difficult to implement and reproduce, but also raises questions about its practical utility compared to simpler approaches. The authors have not adequately demonstrated why such a complicated system is necessary over more straightforward alternatives.

A second major concern is the domain specificity of the approach. While the results in mathematical reasoning are promising, the method appears to be heavily tailored to this particular domain, with multiple components specifically designed for mathematical problem generation. The authors provide no evidence or compelling argument that their approach could generalize to other important domains like code generation or logical reasoning. This significantly limits the broader impact of the work.

The experimental evaluation also has several concerning issues. The comparisons with baselines lack consistency in terms of training data scale, making it difficult to draw meaningful conclusions about the method's effectiveness. Several key results, particularly in the ablation studies, are inadequately explained or justified. The paper fails to provide clear insights into the relative importance of different components in the pipeline, leaving readers uncertain about which elements are truly essential for the method's success.

Furthermore, the selection of different models for different stages of the pipeline appears arbitrary and lacks proper justification. This raises questions about whether the reported improvements are truly attributable to the proposed method rather than simply the careful selection of specific model combinations.

While the goal of making high-quality training data generation more accessible is valuable, and some of the empirical results are intriguing, these fundamental issues with complexity, generalizability, and experimental rigor make it difficult to assess the true value and broader applicability of the proposed method. A more focused approach with clearer justification for design choices, better analysis of component contributions, and demonstration of potential generalizability would be necessary for the work to meet the bar for acceptance. Therefore, I recommend the rejection of this paper in its current form.

**Additional Comments On Reviewer Discussion:**

During the discussion phase, three main concerns were raised by the reviewers. First was the domain specificity of the method, particularly highlighted by reviewers KJ61 and CoxX who questioned whether ScaleQuest could generalize beyond mathematical reasoning. Second was the complexity of the experimental setup and clarity of results, with reviewers FQiU and KJ61 noting issues with baseline comparisons and ablation studies. Third was the general complexity of the pipeline and model selection criteria, raised by all reviewers.

The authors attempted to address these concerns in their response. They extended their evaluation to code reasoning tasks, showing competitive performance against CodeFeedback. They also provided additional ablation studies and analyses, including controlled experiments with equal training data volumes and human evaluations of the generated datasets. The response included clarifications about experimental setup and model selection.

However, these responses don't fully address the fundamental concerns about the method. While the extension to code reasoning is a positive step, it still represents a relatively narrow domain expansion and doesn't demonstrate broader generalizability. The additional analyses, while helpful, highlight rather than resolve the complexity of the approach - revealing even more components and parameters that need careful tuning.

Most importantly, the authors' response reinforces concerns about the method's practical applicability. The need for extensive ablations and analyses suggests that successfully implementing this method requires significant expertise and resources, potentially limiting its practical value to the community despite its theoretical cost advantages over using large proprietary models.

These discussion points ultimately strengthen the case for rejection. While the authors made sincere efforts to address the reviewers' concerns, their responses highlight rather than resolve the fundamental issues with complexity and generalizability that make this work difficult to build upon or adapt to new domains.

---

### Decision · Program_Chairs · 2025-01-22

Reject